# Stochastic Optimal Control for Collective Variable Free Sampling of Molecular Transition Paths

**Lars Holdijk**[*]
University of Oxford

**Yuanqi Du**[*]
AMLab
University of Amsterdam

**Ferry Hooft**
Computational Chemistry Group
University of Amsterdam

**Priyank Jaini**
Google DeepMind

**Bernd Ensing**
AI4Science Lab
Computational Chemistry Group
University of Amsterdam

**Max Welling**
AMLab
University of Amsterdam

## Abstract

We consider the problem of sampling transition paths between two given metastable states of a molecular system, e.g. a folded and unfolded protein or products and reactants of a chemical reaction. Due to the existence of high energy barriers separating the states, these transition paths are unlikely to be sampled with standard Molecular Dynamics (MD) simulation. Traditional methods to augment MD with a bias potential to increase the probability of the transition rely on a dimensionality reduction step based on Collective Variables (CVs). Unfortunately, selecting appropriate CVs requires chemical intuition and traditional methods are therefore not always applicable to larger systems. Additionally, when incorrect CVs are used, the bias potential might not be minimal and bias the system along dimensions irrelevant to the transition. Showing a formal relation between the problem of sampling molecular transition paths, the Schrödinger bridge problem and stochastic optimal control with neural network policies, we propose a machine learning method for sampling said transitions. Unlike previous non-machine learning approaches our method, named PIPS, does not depend on CVs. We show that our method successful generates low energy transitions for Alanine Dipeptide as well as the larger Polyproline and Chignolin proteins.

## 1 Introduction

Molecular Dynamics (MD) is a central tool in the (bio-)chemistry toolbox. By integrating Newton's equations of motion on a molecular scale, MD can provide insight into chemical processes and systems without requiring expensive lab testing [Frenkel and Smit, 2001, Hollingsworth and Dror, 2018]. However, MD is limited when interested in *transitions* between two metastable configurations of a system, such as the folding of a protein, general conformational changes, and chemical reactions. These meta-stable states are separated by regions of high energy which are unlikely to be sampled within a reasonable timespan. While machine learning based approximations of the interatomic forces using neural force fields [Unke et al., 2021] have pushed the boundary in terms of system scale, it does not address the problem of sampling molecular transition paths directly [Fu et al., 2022].

To overcome this issue, prior work in computational and physical chemistry has developed several methods for the enhanced sampling of molecular transitions such as transition path sampling [Bolhuis et al., 2002], umbrella sampling[Torrie and Valleau, 1977] and meta-dynamics [Laio and Parrinello, 2002]. Most of these methods speed up the sampling of transition paths by augmenting the MD simulation with a (learned) bias potential that pushes the system to cross the energy barrier separating two states. However, due to the large configuration space of molecular trajectories, finding such a bias potential is in itself a computationally expensive task.

37th Conference on Neural Information Processing Systems (NeurIPS 2023).

To circumvent this problem, prior methods depend on *Collective Variables* (CVs). CVs are functions of atomic coordinates that have been identified as playing a role within the transition period. Biasing methods rely on these CVs to reduce the complexity of the bias potential by only biasing the system along them. Limiting the bias potential to act on the CVs is an intuitive approach since the most common reason to sample transition paths, deriving transition dependent quantities such as reaction free-energy and reaction rate, are functions of CVs themselves [Bussi and Branduardi, 2015]. See fig. 1 for an illustration of the free-energy barrier separating two metastable states of the Alanine Dipeptide protein for which the dihedral angles $\phi$ and $\psi$ are known to be CVs.

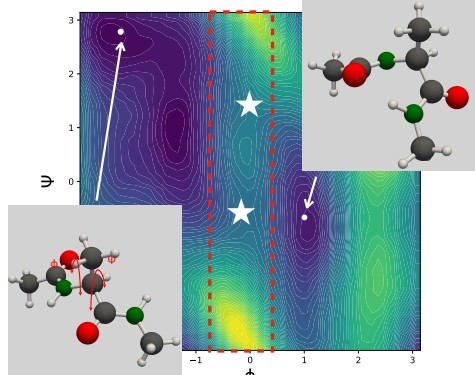

Figure 1: Free-energy surface of Alanine Dipeptide as a function of CV dihedral angles $\phi$ and $\psi$ highlighting the high energy barrier separating the two metastable states. White stars indicate saddle points in the high energy barrier where the transition is likely to occur.

However, while sensible, depending on CVs to reduce the dimensitionality of bias potential search space is not always suitable. While some methodological approaches are available [Hooft et al., 2021] for smaller systems, selecting CVs relies on prior expert knowledge. This limits the applicability of bias potential enhanced sampling to systems for which this information is available. Additionally, when CVs are incorrectly chosen, depending the bias potential on these CVs might result in errors in determining dependent quantities [Bolhuis et al., 2000] and incorrect interpretation of the transition process.

For this purpose, we propose PIPS, a Path Integral stochastic optimal control [Kappen, 2005, Kappen and Ruiz, 2016] method for Path Sampling of molecular transitions. PIPS leverages stochastic optimal control theory to train a parameterised bias potential that, unlike previous methods from computational chemistry, operates on the entire geometry of the molecule instead of depending on predetermined CVs. This way, PIPS can be scaled to larger systems.

**Contributions and outline** Our contributions are organised as follows. First, we introduce the problem of sampling transition paths in section 2. Second, we formally show in section 3 the relationship between the problem of sampling transitions paths, the Schrodinger Bridge Problem (section 3.1) and Stochastic Optimal Control (SOC) (section 3.3). Following this, we use the gained insights regarding SOC in section 4 to propose PIPS; a method based on the PICE algorithm designed for sampling molecular transition paths that does not depend on Collective Variables. Lastly, we demonstrate the efficacy of PIPS on conformational transitions in three molecular systems of varying complexity, namely Alanine Dipeptide, Polyproline, and Chignolin in section 5.

## 2 Preliminaries, Problem Setup, and Related Work

### 2.1 Molecular Dynamics

Given the state of a molecular system $\boldsymbol{x}_t := (\boldsymbol{r}_t, \boldsymbol{v}_t)$ consisting of positions $\boldsymbol{r}_t \in \mathbb{R}^{3n}$ and velocities $\boldsymbol{v}_t \in \mathbb{R}^{3n}$ at time $t$ with $n$ atoms sampled from the Gibbs distribution $\pi_G(\boldsymbol{x}_t) = \exp(-\frac{1}{k_B T}\mathcal{H}(\boldsymbol{r}_t, \boldsymbol{v}_t))$, Molecular Dynamics (MD) describe the time evolution of the state over time. $\mathcal{H}$ is known as the Hamiltonian $\mathcal{H}(\boldsymbol{r}_t, \boldsymbol{v}_t) = U(\boldsymbol{r}_t) + K(\boldsymbol{v}_t)$, where $U(\boldsymbol{r}_t)$ and $K(\boldsymbol{v}_t) = \frac{1}{2}\boldsymbol{m}\boldsymbol{v}^2$, with mass $\boldsymbol{m}$, respectively denote the Potential and Kinetic Energy of the system. The potential energy of a system is defined by a parameterized sum of pairwise empirical potential functions, such as harmonic bonds, angle potentials, inter-molecular electrostatic and Van der Waals potentials.

One common approach of integrating the molecular dynamics is Langevin Dynamics [Bussi and Parrinello, 2007] which couple the deterministic Newtonian equations of motion with a stochastic thermostat that acts like a heat bath. Langevin dynamics obey the following SDEs

$$\mathrm{d}\boldsymbol{r} = \boldsymbol{v} \cdot \mathrm{d}t \tag{1}$$

$$\mathrm{d}\boldsymbol{v} = \frac{-\nabla_{\boldsymbol{r}} U(\boldsymbol{r})}{\boldsymbol{m}} \cdot \mathrm{d}t - \gamma \boldsymbol{v} \cdot \mathrm{d}t + \sqrt{2\boldsymbol{m}\gamma k_B T}\,\mathrm{d}\boldsymbol{W}, \tag{2}$$

where $k_B$ is the Boltzmann constant, $T$ the temperature of the heath bath, and $\mathrm{d}\boldsymbol{W}$ standard Brownian Motion. $\gamma$, the friction term, couples the dynamics and the heat bath. Following this SDE samples samples from the Canonical of NVT ensemble with constant temperature.

## 2.2 Sampling Transition Path Sampling

By sampling an initial configuration $\boldsymbol{x}_0 = (\boldsymbol{r}_0, \boldsymbol{v}_0) \sim \pi_G$ and following the MD simulation for a fixed amount of time, one can generate trajectories $\boldsymbol{x}_{0:\tau} = \{\boldsymbol{x}_0, \ldots, \boldsymbol{x}_\tau\}$, of length $\tau$. These trajectories represent samples from the probability distribution over trajectories given by:

$$\pi(\boldsymbol{x}_{0:\tau}) = \pi_G(\boldsymbol{x}_0) \cdot \prod_{t=1}^{\tau} \mathcal{N}(\boldsymbol{x}_t | \boldsymbol{\mu}_{t-1}, \boldsymbol{\Sigma}_{t-1}), \tag{3}$$

with $\boldsymbol{\mu}_t = (\boldsymbol{v}_t \cdot \mathrm{d}t, \frac{-\nabla_r U(\boldsymbol{r}_t)}{\boldsymbol{m}} \cdot \mathrm{d}t - \gamma \boldsymbol{v}_t \cdot \mathrm{d}t)^T$ and $\boldsymbol{\Sigma} = \mathrm{diag}(0, 2\boldsymbol{m}\gamma k_B T)$.

However, in the context of sampling transition paths, we are interested in trajectories with a predefined an initial and final state. Ie. $\boldsymbol{r}_0 \in R \subset \mathbb{R}^{3n}$ and $\boldsymbol{r}_\tau \in P \subset \mathbb{R}^{3n}$. For example, $R$ can describe the set of reactants and $P$ the set of products of a chemical reaction. Or, $R$ can be the set of stable native states of a protein while $P$ is the set of folded proteins.

We will refer to the distribution over trajectories with restricted initial and target states as the *Transition Path* (TP) distribution [Dellago et al., 1998].

**Definition 1** (Transition Path (TP) distribution). *Given a set of initial states $R$, target states $P$, potential energy $U$ and a transition length $\tau$ the Transition Path (TP) distribution is defined as;*

$$\pi^*(\boldsymbol{x}_{0:\tau}) = \frac{1}{Z} \mathbf{1}_R(\boldsymbol{r}_0) \cdot \pi(\boldsymbol{x}_{0:\tau}) \cdot \mathbf{1}_P(\boldsymbol{r}_\tau) \tag{4}$$

*where $\mathbf{1}_R$ and $\mathbf{1}_P$ are indicator functions and $\pi(\boldsymbol{x}_{0:\tau})$ is defined according to eq. (3).*

We can naively apply rejection sampling to sample from the TP distribution by sampling a system $\boldsymbol{x}_0 \sim \mathbf{1}_R(\boldsymbol{r}_0) \cdot \pi_G(\boldsymbol{x}_0)$, evolving it for $\tau$ steps according to the MD in eq. (1) and rejecting it when $\boldsymbol{r}_\tau \notin P$. Unfortunately, when using standard molecular dynamics, it is very unlikely for any trajectory starting in a state $\boldsymbol{r}_0 \in R$ to terminate with $\boldsymbol{r}_\tau \in P$ due to the two sets of states being separated by a high-energy barrier. Ie. for all $\boldsymbol{x}_{0:\tau} \sim \pi^*$ some $\boldsymbol{x}_t$ has $U(\boldsymbol{r}_t) >> U(\boldsymbol{r}_0)$. To be able to obtain a representative number of trajectories, one is thus forced to generate a high number of trajectories, making naive sampling from the TP distribution computationally very expensive.

## 2.3 Bias Potential Enhanced Sampling

To solve the problem caused by high-free energy barriers and to sample from the TP distribution various enhanced sampling approaches are available. These will be further discussed in the related work section. In this work, we will focus on a specific branch of enhanced sampling methods called *Bias Potential Enhanced Sampling* (BPES). In BPES approaches, the stochastic dynamics are enhanced with a bias potential $b(\boldsymbol{r}, \boldsymbol{v})$ such that when a system $\boldsymbol{x}_0 \sim \mathbf{1}_R(\boldsymbol{r}_0) \cdot \pi_G(\boldsymbol{x}_0)$ is transformed according to the biased dynamics

$$\mathrm{d}\boldsymbol{r} = \boldsymbol{v} \cdot \mathrm{d}t \tag{5}$$

$$\mathrm{d}\boldsymbol{v} = \frac{-\nabla_{\boldsymbol{r}}\big(U(\boldsymbol{r}) + b(\boldsymbol{r}, \boldsymbol{v})\big)}{\boldsymbol{m}} \cdot \mathrm{d}t - \gamma \boldsymbol{v} \cdot \mathrm{d}t + \sqrt{2\boldsymbol{m}\gamma k_B T}\, \mathrm{d}\boldsymbol{W}, \tag{6}$$

a trajectory, of length $\tau$, always terminates with $\boldsymbol{r}_\tau \in P$.

Trajectories sampled by following these bias potential enhanced dynamics are sampled according to what we refer to as the Bias Potential enhanced Transition Path (BPTP) distribution

$$\pi^b(\boldsymbol{x}_{0:\tau}) = \mathbf{1}_R(\boldsymbol{r}_0) \cdot \pi_G(\boldsymbol{x}_0) \cdot \prod_{t=1}^{\tau} \mathcal{N}(\boldsymbol{x}_t | \hat{\boldsymbol{\mu}}_{t-1}, \hat{\Sigma}_{t-1}), \tag{7}$$

with $\hat{\boldsymbol{\mu}}_t = (\boldsymbol{v}_t \cdot \mathrm{d}t, \frac{-\nabla_r\big(U(\boldsymbol{r}_t) + b(\boldsymbol{r}_t, \boldsymbol{v}_t)\big)}{\boldsymbol{m}} \cdot \mathrm{d}t - \gamma \boldsymbol{v}_t \cdot \mathrm{d}t)^T$ and $\hat{\Sigma} = \mathrm{diag}(0, 2\boldsymbol{m}\gamma k_B T)$.

Finding the bias potential $b(\boldsymbol{r}, \boldsymbol{v})$ such that trajectories sampled from the BPTP distribution are distributed according to the TP distribution is referred to as the BPTP problem.

**Definition 2** (BPTP problem). *Given a set of initial states $R$, target states $P$ and a Potential Energy function $U$, the BPTP problem describes the task of finding an optimal bias potential $b^*$ such that trajectories sampled from the BPTP distribution $\pi^{b^*}$ are close to samples sampled to the TP distribution $\pi^*$, ie.*

$$b^* = \arg\min_b \mathbb{D}_{\mathsf{KL}}(\pi^b | \pi^*). \tag{8}$$

### 2.3.1 Related Enhanced Sampling Methods

**CV dependent Enhanced Sampling** Most closely related to our work are the metadynamics [Laio and Parrinello, 2002, Bussi and Branduardi, 2015, Barducci et al., 2008] and the Adaptive Biasing Force (ABF) methods [Darve and Pohorille, 2001, Comer et al., 2015]. In metadynamics, the bias potential is iteratively built as a sum of Gaussians centered at conformational states previously visited during the MD simulation. This consecutively pushes the system outwards to regions of higher energy not previously visited. Contrarily to metadynamics, ABF does not aim to learn the bias potential $b(\boldsymbol{r}, \boldsymbol{v})$, but instead aims to control the system through the *bias force* $\boldsymbol{b}(\boldsymbol{r}, \boldsymbol{v}) = \nabla_{\boldsymbol{r}} b(\boldsymbol{r}, \boldsymbol{v}) \in \mathbb{R}^{3n}$. The intuition behind ABF is to learn a bias force that cancels out the deterministic force from the molecular potential. As a result, the only remaining driving force is the stochastic Langevin thermostat which is not affected by the high energy barriers. Other approaches to sampling transition paths using a bias potential include umbrella sampling [Torrie and Valleau [1977], hyper-MD [Voter, 1997], the Wang-Landau method [Wang and Landau, 2001] and various less commonly applied others [Sprik and Ciccotti, 1998, Grubmüller, 1995, Huber et al., 1994, Carter et al., 1989]. All these methods depend on dimensionality reduction steps using CVs while our proposed method does not.

**CV free Enhanced Sampling** In addition to the CV dependent methods a different family of MCMC based approaches for direct sampling from the TP distributions is available. These methods, such as Transition Path Sampling [Dellago et al., 1998, Bolhuis et al., 2002] and Nudge Elastic Band sampling [Henkelman et al., 2000], generally do not use a bias potential or CVs.

Recently, several machine learning solutions for the BPTP and related problems have been proposed. For example, Das et al. [2021] use Reinforcement Learning to sample from the TP distribution under Brownian dynamics, Schneider et al. [2017] consider the modelling of the free-energy surface using neural networks, and both Sultan et al. [2018] and Sun et al. [2022] use neural networks find CVs.

## 3 Sampling Transition Paths using Stochastic Optimal Control theory

In this section we will formally discuss the relationship between the BPTP problem and two topics from the machine learning literature; the Schrodinger Bridge problem and Stochastic Optimal Control.

### 3.1 The BPTP problem is a Schrodinger Bridge Problem

First introduced by Schrodinger [Schrödinger, 1931, 1932], the Schrodinger Bridge (SB) problem studies the transition between two distributions over time under some fixed drift and diffusion components. Formally, the SP problem is defined as

**Definition 3** (Schrodinger Bridge (SB) problem). *Given a reference distribution* $\pi\big(\boldsymbol{x}_{0:\tau}\big)$ *over trajectories with predefined marginals* $\pi_0$ *and* $\pi_\tau$*, the Schrodinger Bridge (SB) Problem aims to find an alternative distribution* $\hat{\pi}\big(\boldsymbol{x}_{0:\tau}\big)$ *such that*

$$\hat{\pi}^*\big(\boldsymbol{x}_{0:\tau}\big) := \underset{\hat{\pi}(\boldsymbol{x}_{0:\tau}) \in \mathcal{D}(\pi_0, \pi_\tau)}{\arg\min} \mathbb{D}_{\mathsf{KL}}\Big(\hat{\pi}\big(\boldsymbol{x}_{0:\tau}\big) \| \pi\big(\boldsymbol{x}_{0:\tau}\big)\Big) \tag{9}$$

*where* $\mathcal{D}(\pi_0, \pi_\tau)$ *is the space of path measures with marginals* $\pi_0$ *and* $\pi_\tau$.

Recently, machine learning approaches for parameterizing this alternative distribution $\hat{\pi}$ to approximate the reference distribution $\pi$ have received attention [Vargas et al., 2021, De Bortoli et al., 2021]. In the following theorem, we show that these approaches also provide a solution to the BPTP problem when the correct marginal distributions are specified.

**Theorem 3.1** (BPTP problem is a SB problem). *Let* $b$ *be the set of functions such that* $\pi_0 = \pi_G(\boldsymbol{x}_0) \cdot \mathbf{1}_R(\boldsymbol{r}_0)$ *and* $\pi_\tau = \pi_G(\boldsymbol{x}_\tau) \cdot \mathbf{1}_R(\boldsymbol{r}_\tau)$*, we have that a solution to the SB problem with reference distribution* $\pi^*$ *is also a solution to the BPTP problem, ie.*

$$\underset{b}{\arg\min} \, \mathbb{D}_{\mathsf{KL}}(\pi^b | \pi^*) = \underset{\pi^b \in \mathcal{D}(\pi_0, \pi_\tau)}{\arg\min} \mathbb{D}_{\mathsf{KL}}\Big(\pi^b \| \pi^*\Big) \tag{10}$$

*Proof.* This follows from the definition of the BPTP and SB problems. $\square$

Following this theorem, we can use proposed solutions for solving the SBP to solve the BPTP problem using a bias potential. In this work, we will specifically focus on Stochastic Optimal Control theory, which has been shown to solve the SBP in [Chen et al., 2016].

## 3.2 Background: Stochastic Optimal Control

Before formally introducing the relation between Stochastic Optimal Control (SOC) and the BPTP problem we first review some of the basic concepts of SOC and, more specifically, the Path Integral Stochastic Optimal Control (PISOC) branch of SOC theory.

Given an arbitrarily controlled dynamical system

$$\mathrm{d}\boldsymbol{x}_t = \boldsymbol{f}(\boldsymbol{x}_t)\,\mathrm{d}t + \boldsymbol{G}(\boldsymbol{x}_t)\cdot\big(\boldsymbol{u}(\boldsymbol{x}_t)\,\mathrm{d}t + \mathrm{d}\boldsymbol{W}\big), \qquad \boldsymbol{x}_0 \sim \pi_0, \tag{11}$$

where $\boldsymbol{f} : \mathbb{R}^d \times \mathbb{R}^+ \to \mathbb{R}^d$ and $\boldsymbol{G} : \mathbb{R}^d \times \mathbb{R}^+ \to \mathbb{R}^{d\times d}$ are deterministic functions representing the drift and volatility of the system and $\mathrm{d}\boldsymbol{W}$ is Brownian Motion with variance $\nu$, Stochastic Optimal Control theory aims to find a policy $\boldsymbol{u}(\boldsymbol{x}_t)$ that minimizes some expected cost $C$ over the trajectories:

$$\boldsymbol{u}^* = \arg\min_{\boldsymbol{u}} \mathbb{E}_{\boldsymbol{x}_{0:\tau}\sim\pi_u}\big[C(x_{0:\tau})\big] \tag{12}$$

Here $\pi_u$ represents the distribution over trajectories similar to eq. (7) with $\boldsymbol{\mu}_t = \boldsymbol{x}_t + \boldsymbol{f}(\boldsymbol{x}_t, t)\,\mathrm{d}t + \boldsymbol{G}(\boldsymbol{x}_t)(\boldsymbol{u}(\boldsymbol{x}_t)\,\mathrm{d}t)$ and $\Sigma_t = \boldsymbol{G}(\boldsymbol{x}_t)^T\nu\boldsymbol{G}(\boldsymbol{x}_t)$.

In this work we will specifically rely on a branch of SOC called Path Integral Control (PISOC), first introduced by Kappen [2007]. In PISOC the cost of a trajectory is defined as

$$C(\boldsymbol{x}_{0:\tau}) = \frac{1}{\lambda}\Big(\varphi(\boldsymbol{x}_\tau) + \sum_{t=0}^{\tau-1}\frac{1}{2}\boldsymbol{u}(\boldsymbol{x}_t)^T\boldsymbol{R}\boldsymbol{u}(\boldsymbol{x}_t) + \boldsymbol{u}(\boldsymbol{x}_t)^T\boldsymbol{R}\boldsymbol{\varepsilon}_t\Big) \tag{13}$$

where $\boldsymbol{\varepsilon}_t = \boldsymbol{G}^{-1}(\boldsymbol{x}_t)(\mathrm{d}\boldsymbol{x} - \boldsymbol{f}(\boldsymbol{x}_t)\,\mathrm{d}t) - \boldsymbol{u}(\boldsymbol{x}_t)$, $\varphi$ denotes the terminal cost, $\lambda$ is a constant and $\boldsymbol{R}$ is the cost of taking action $\boldsymbol{u}$ in the current state and is given as a weight matrix for a quadratic control cost. To clarify, $\boldsymbol{\varepsilon}_t \sim \mathrm{d}\boldsymbol{W}$ is the noise introduced into the trajectories by the Langevin thermostat.

The last term in the cost function in eq. (13) relating the Brownian motion and the control is unusual and devoid of a clear intuition. However, this term plays an important role when relating the cost to a KL-divergence which we will establish next. Additionally, as discussed by Thijssen and Kappen [2015], the additional cost vanishes under expectation ($\mathbb{E}_{\boldsymbol{x}_{0:\tau}\sim\pi_u}[\boldsymbol{u}(\boldsymbol{x}_t)^T\boldsymbol{R}\boldsymbol{\varepsilon}_t] = 0$) and thus, does not influence the optimal control $\boldsymbol{u}^*$ given by eq. (12).

## 3.3 Stochastic Optimal Control solves the BPTP problem

We can see that SOC dynamical system (eq. (11)) is similar to the dynamics of the BPTP distribution (eq. (5)). In fact, as we will see next, with a properly defined $\varphi$, minimizing the trajectory cost (eq. (13)) results in finding a control $\boldsymbol{u}$ that solves the BPTP problem.

**Theorem 3.2** (SOC solves the BPTP problem). *Given $\boldsymbol{x}_t = (\boldsymbol{r}_t, \boldsymbol{v}_t)^T$, $\boldsymbol{f}(\boldsymbol{x}_t) = (\boldsymbol{v}_t, \frac{-\nabla_{\boldsymbol{r}_t}U(\boldsymbol{r}_t)}{\boldsymbol{m}} - \gamma\boldsymbol{v}_t)^T$, $\boldsymbol{G}(\boldsymbol{x}_t) = (\boldsymbol{0}_{3n}, \mathbb{I}_{3n})^T$, $\boldsymbol{u}(\boldsymbol{x}_t) = \frac{-\nabla_{\boldsymbol{r}_t}b_\theta(\boldsymbol{r}_t, \boldsymbol{v}_t)}{\boldsymbol{m}}$, $\nu = 2\boldsymbol{m}\gamma k_B T$, and $\pi_0 = \pi_G$, such that the SOC dynamics (eq. (11)) describe the dynamics of the BPTP distribution $\pi^b$ (eq. (5)).*

*If we define $\varphi(\boldsymbol{x}_\tau) = -\lambda\log(\boldsymbol{1}_P(\boldsymbol{r}_\tau))$, $\boldsymbol{R} = \lambda\nu^{-1} = \lambda(2\boldsymbol{m}\gamma k_B T)^{-1}$ and assume $\boldsymbol{r}_0 \in R$, we have*

$$\arg\min_b \mathbb{E}_{\boldsymbol{x}_{0:\tau}\sim\pi^b}\big[C(\boldsymbol{x}_{0:\tau})\big] = \arg\min_b \mathbb{D}_{\mathsf{KL}}(\pi^b|\pi^*), \tag{14}$$

*where $\pi^*$ is the TP distribution.*

*Proof.* See appendix A. The proof relates $\pi^b$ and $\pi^0$ using Girsanov's theorem to rewrite the expectation over cost $C$ as the summation of the terminal cost and a KL divergence. $\qquad\square$

Using the established connection we now thus have a tool to solve the BPTP problem by learning a parameterized control $\boldsymbol{u}_\theta(\boldsymbol{x}_t) = \frac{-\nabla_{\boldsymbol{r}_t}b_\theta(\boldsymbol{r}_t, \boldsymbol{v}_t)}{\boldsymbol{m}}$ and consequentially the parameterized bias potential $b_\theta$ using SOC theory. In the following section we will look at one specific approach for doing so.

# 4 PIPS: Path Integral SOC for Path Sampling

Following the formal construction of the relationship between SOC and the BPTP problem, we now introduce our proposed method to find the parameterized bias potential $b_\theta$ based on this connection. We refer to this method as PIPS: Path Integral Path Sampling. PIPS is an adaptation of the Path Integral Cross Entropy (PICE) [Kappen and Ruiz, 2016] method to the setting of sampling molecular transition paths where we have a single initial $R = \{r_0^*\}$ and target $P = \{r_\tau^*\}$ system.

**Background: Path Integral Cross Entropy**  Kappen and Ruiz [2016] introduced the Path Integral Cross Entropy (PICE) method for solving Equation (12). The PICE method derives an explicit expression for the distribution $\pi^{u^*}$ under optimal control $u^*$ when $\lambda = \nu R$ given by:

$$\pi^{u^*} = \frac{1}{\eta(x, t)} \pi^u (x_{0:\tau}) \exp(-C(x_{0:\tau})) \tag{15}$$

where $\eta(\tau) = \mathbb{E}_{x_{0:\tau} \sim \pi^0}[\exp(-\frac{1}{\lambda}\varphi(x_\tau)]$ is the normalization constant. This establishes the optimal distribution $\pi^{u^*}$ as a reweighing of any distribution induced by an arbitrary control $u$.

PICE, subsequently, achieves this by minimizing the KL-divergence between the optimal controlled distribution $\pi^{u^*}$ and a parameterized distribution $\pi^{u_\theta}$ using gradient descent as follows:

$$\frac{\partial \mathbb{D}_{\mathsf{KL}}(\pi^{u^*}|\pi^{u_\theta})}{\partial \theta} = -\frac{1}{\eta}\mathbb{E}_{x_{0:\tau} \sim \pi_{u_\theta}}[\exp(-C(x_{0:\tau}, u_\theta)) \sum_{t=0}^{\tau}(R\varepsilon_t \cdot \frac{\partial u_\theta}{\partial \theta})] \tag{16}$$

Similar to the optimal control in eq. (15), the gradient used to minimize the KL-divergence is found by reweighing for each sampled trajectory, $x_{0:\tau}$, the gradient of the control policy $u_\theta$ by the cost of the trajectory. See Algorithm 1 in the appendix for an algorithmic description of PICE.

## 4.1 Adaptations to PICE

In this section we will specify the adaptations made to the PICE algorithm to apply it to solve the BPTP problem for the molecular transition path setting.

**Smoothing the loss function**  As shown in the previous section, when using the target loss $\varphi(x_\tau) = -\lambda \log(\mathbf{1}_P(r_\tau))$, SOC solves the BPTP problem. However, while optimal, this loss function is hard to use in the PICE optimization task as it is infinite for all $x_{0:\tau}$ where $r_\tau \neq r_\tau^*$. As such, we instead use a smoothed version $\varphi(r_t) = \exp \sum_{i,j}^n (d_{ij}(r_t) - d_{ij}(r_\tau))^2$ where $d_{ij}(r_t) = \|(r_t)_i - (r_t)_j\|_2^2$. This exponentiated pairwise distance between the atoms is a commonly used distance metric [Shi et al., 2021] that is invariant to rotations and translations of the molecular system.

**Architectural considerations**  The learnable component of PIPS is the bias potential $b$. However, as the BPTP dynamics show in eq. (5), instead of using the bias potential directly, MD depends on the *bias force* — the gradient of the bias potential $b(r, v) = \nabla_r b(r, v) \in \mathbb{R}^{3n}$. This consideration allows for two different modelling approaches for the bias force similar to the distinction between metadynamics and adaptive bias force discussed in section 2.3.1. One can either parameterise the bias force directly $b(r, v) = b_\theta(r, v)$ or , alternatively, model $b_\theta(r, v)$ the bias potential and calculate the corresponding bias force by backpropagation, $b(r, v) = \nabla_r b_\theta(r, v)$. The advantage of the latter is that the forces are conservative by construction.

In section 5.1 we will compare both these modelling approaches. In both cases we will use a MLP with ReLU activation for either the parameterized bias force or bias potential. Alternatively, $b_\theta$ or $b_\theta$ could be implemented using recent advances in physics inspired equivariant neural networks [Cohen and Welling, 2016, Satorras et al., 2021] that take into account the SE(3) symmetry of the system. We provide details for training the control network $u_\theta$ in Appendix B.

**Integration with MD simulation frameworks**  To efficiently calculate the Potential $U(x)$ and integrate the MD in eq. (1), various optimized simulation frameworks are available. In our work we use the OpenMM framework [Eastman et al., 2017]. The bias force $b(r, v)$ is integrated in OpenMM as a *custom external force*. Implementing the control this way allows us to use the

optimized configuration capabilities of OpenMM, such as forcefield definitions (the potential function description) and integrators (for the time-discretization of our dynamics).

One downside of using OpenMM for integrating the MD is that it does not provide access to the noise $\varepsilon_t \sim \sqrt{2m\gamma k_B T}\,\mathrm{d}\boldsymbol{W}$ used in the Langevin thermostat that is needed to calculate the update to the policy weights. To circumvent this, we instead sample an additional exploratory noise term $\hat{\varepsilon}_t \sim \mathrm{d}\boldsymbol{W}$ with variance $\hat{\nu}$ that is used to optimize the policy and assume the Langevin noise to be part of the drift of the system $\boldsymbol{f}$. While this loses the formal guarantees presented in section 3, we found this to be experimentally stable and provide close to optimal trajectory paths (as shown in section 5.1).

## 5 Experiments

We evaluate PIPS using three molecular systems, namely (i) **Alanine Dipeptide**, to compare PIPS to CV free and CV dependent baselines, (ii) **Polyproline**, to evaluate PIPS as a method to select candidate CVs, and (iii) **Chignolin**, as a use-case of PIPS scalabilty to proteins without knowns CVs.

We report the molecule specific OpenMM configuration as well as the used neural network architecture to learn the bias potential/force in appendix C. Generally, we run our simulations at 300 K and use 6 layer MLP with the width of the layers dependent on the number of atoms in the molecule under consideration. Our code, including a full stand-alone notebook re-implementation, is available here: https://github.com/LarsHoldijk/SOCTransitionPaths.

### 5.1 Alanine Dipeptide

In this section we evaluate PIPS on the extensive studied Alanine Dipeptide (AD) molecule. AD is a relatively small protein for which the CVs (two dihedral angles $\phi$ and $\psi$) are readily available and is therefore well suited for the development of enhanced sampling methods that require CVs. While PIPS does not use the CVs during training, their availability does come in useful to evaluate the sampled transition. The transition evaluated here have a 500 fs time horizon.

#### 5.1.1 Quantitative comparison to CV free baselines

As discussed, our work is the first to consider CV free sampling of transition paths at this scale and as such other baselines or metrics are not available. In table 1 we therefore evaluate PIPS using MD simulations with extended time-horizon and increased system temperature as baselines and introduce three metrics to evaluate the quality of the transition paths. (i) *Expected Pairwise Distance* (EPD) measures the euclidean distance between the final conformation in the trajectory and the target conformation, reflecting the goal of the transition to end in the target state, (ii) *Target Hit Percentage* (THP) assures that the final configuration is also close in terms of CVs by measuring the percentage of trajectories correctly transforming these CVs, and (iii) *Energy Transition Point* (ETP)

| | $\tau$ fs | Temp. K | EPD ($\downarrow$) nm $\times 10^{-3}$ | THP ($\uparrow$) % | ETP ($\downarrow$) kJ mol$^{-1}$ |
|---|---|---|---|---|---|
| Bias Force Prediction | 500 | 300 | 2.56 ± 0.34 | 45.0 % | 0.55 ± 11.30 |
| Bias Potential Prediction | 500 | 300 | 1.21 ± 0.31 | 63.5 % | -8.35 ± 8.04 |
| MD w. fixed timescale | 500 | 300 | 8.50 ± 0.67 | 0% | - |
| | 500 | 1500 | 7.75 ± 1.72 | 0% | - |
| | 500 | 4500 | 6.77 ± 2.41 | 0.1% | 317.79 ± 0.00 |
| | 500 | 9000 | 6.99 ± 2.56 | 1.6 % | 772.57 ± 108.55 |
| MD w/ fixed timescale | 6818.4 ± 5420.8 | 1500 | 3.08 ± 0.68 | 100% | 393.76 ± 68.67 |
| | 3471.7 ± 1646.5 | 4500 | 6.42 ± 2.67 | 100% | 1186.84 ± 212.00 |

Table 1: Benchmark scores for the proposed method and extended MD baselines. From-left-to-right: Time-horizon $\tau$ representing the trajectory length (note that we take one policy step every 1 fs), simulation temperature, Expected Pairwise distance (EPD), Target Hit Percentage (THP), and Energy Transition Point (ETP). ETP can only be calculate when a trajectory reaches the target. All metrics are averaged over 1000 trajectories except for MD w/ fixed timescale which is ran only for 10 trajectories.

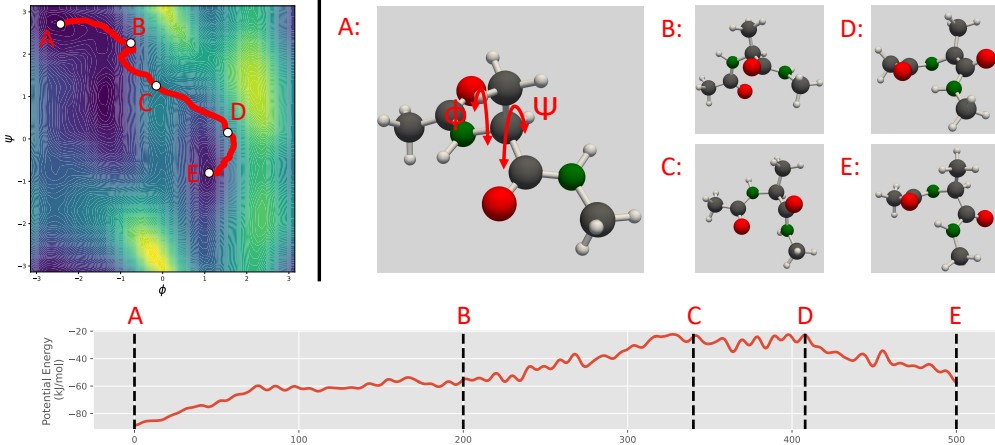

Figure 2: Visualization of a trajectory sampled with PIPS. *Left:* The sampled trajectory projected on the free energy landscape of AD as a function of two CVs *Right:* Conformations along the sampled trajectory: A) starting conformation showing the CV dihedral angles, B-D) intermediate conformations with C being the highest energy point on the trajectory, and E) final conformation, which closely aligns with the target conformation. *Bottom:* Potential energy during transition.

which evaluates the capacity of each method to find transition paths that cross the high-energy barrier at a low point by taking the maximum potential energy of the molecule along the trajectory. A good trajectory will be one that passes through the minimal high-energy barrier and ETP aims to measure this. We provide more details in Appendix C.2.1.

**Results:** We find that the trajectories generated by both the policy networks outperform the MD baselines, but the more physics-aligned potential predicting policy performs best under our metrics. This policy network consistently reaches the target conformation both in terms of full geometry and the CVs orientation. Furthermore, our policy network generates these trajectories in a significantly shorter time than temperature enhanced MD simulations without a fixed timescale. When we do limit MD to run for the same timescale as the proposed method, we found that, in contrast to the proposed method, temperature enhanced MD simulations are unable to generate successful trajectories. We will use the bias potential predicting policy in the following.

### 5.1.2 Qualitative comparison to CV dependent metadynamics

In fig. 2 we visualise an AD transition sampled by PIPS using the bias potential predicting policy. In the top left, we overlay the transition projected onto CV space on the free-energy surface generated using metadynamics. The free-energy surface thus serves as a ground-truth generated using a method that requires extensive domain knowledge. We aim to show that the trajectory sampled using PIPS aligns with the saddle points of the metadynamics free-energy surface.

**Results:** The trajectory in Figure 2 demonstrates that the bias potential control policy transforms the molecule from the initial position (A) to the final position (E) by transitioning over the same saddle point in the free-energy barrier found by metadynamics (C). This shows that the trajectory follows the same transition in CV space as metadynamics despite, contrarily to metadynamics, not being biased to do so. The potential energy goes up during the transition until it reaches the lowest point of the energy barrier (C) and consecutively settles down in its new low-energy state.

### 5.2 Polyproline Helix

Second, we consider a Polyproline trimer with 3 proline residues. Polyproline is a more complex protein then AD and as such its CVs are less well understood. We therefore use this protein to determine if PIPS biases the system along the correct CVs when a collection of candidate CVs are available. Specifically, we consider the peptide bond orientation ($\omega$) and two backbone dihedral angles ($\phi$ and $\psi$). As initial and target state we provide a single example of Polyprolines PP-I form (with cis-isomer peptide bonds) and PP-II form (with trans-isomer peptide bonds) respectively. For this transition it is known that the CV of interest are the peptide bond orientation. Additionally, to

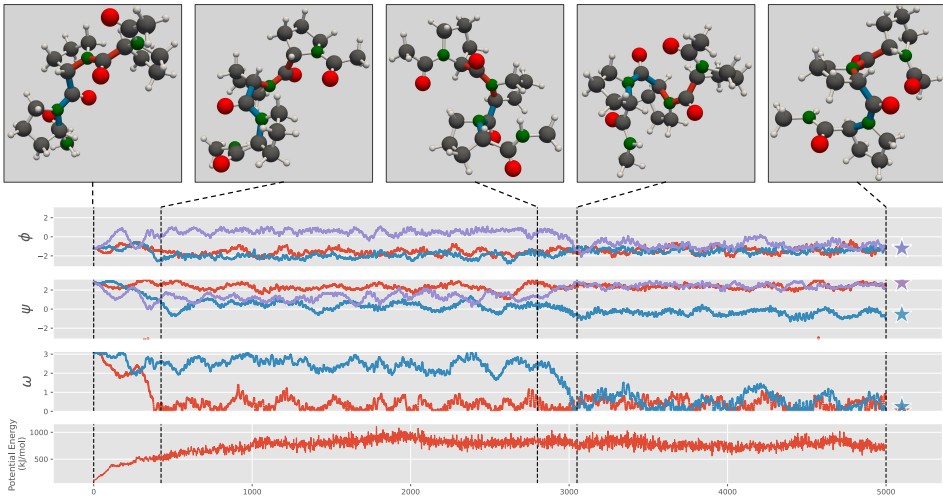

Figure 3: Visualization of the Polyproline transformation from PP-II to PP-I. *From-top-to-bottom* 5 stages of the transition, $\psi$, $\phi$, $\omega$ candidate CVs, and Potential Energy. For the candidate CVs multiple instances of the same dihedral angles can be found in a single molecule. Stars indicate target candidate CV states. Colored bonds represent the bonds involved in the $\omega$ CV.

study PIPS resilience to target misspecification, the supplied PP-II form also contains a transformation in one of the $\psi$-dihedral angles which is irrelevant to the transition. The transition time is $5000\,\text{fs}$.

**Results:** We visualize the transformation of the three collective variables $(\omega, \phi, \psi)$ as well as the corresponding potential energy of the conformation in Figure 3 for a sampled transition path. We observe that the transition correctly occurs along the $\omega$ CV going from $180°$ to $0°$. This suggest that PIPS could be used for testing the validity of candidate CVs. However, we also observe that in addition to the peptide bonds PIPS also biases the system along one of the $\psi$-dihedral angles due to the introduced target misspecification. As the small fluctuations are to be expected when sampling a single target from the Boltzmann distribution, alternative methods for specifying the target state should be explored in future work.

### 5.3   Chignolin

Lastly, we consider the small $\beta$-hairpin protein Chignolin. Chignolin was artificially constructed to study protein folding mechanisms [Honda et al., 2004, Seibert et al., 2005]. Due to its small size, its folding process is easier to study than larger scale proteins while being similar enough to shed light on this complex process. In contrast to Alanine Dipeptide and Polyproline, there is no agreement on the transition mechanism describing the folding of Chignolin. Both the CVs involved [Satoh et al., 2006, Paissoni and Camilloni, 2021], as well as the sequence of steps [Harada and Kitao, 2011, Satoh et al., 2006, Suenaga et al., 2007, Enemark et al., 2012] describing the folding process have multiple different interpretations. Chignolin thus serves as a usecase study for scaling PIPS beyond traditional CV-based approaches to solve the BPTP-problem. We sample transition paths between the folded and unfolded state of the Chignolin protein using a total time horizon of $5000\,\text{fs}$. Note that the typical folding time of Chignolin is recorded to be $0.6\,\mu\text{s}$ [Lindorff-Larsen et al., 2011].

**Results:** In Figure 4, we visualize the transition of Chignolin at 5 different timesteps during the transition path. We observe that to transition the protein from its low energy unfolded state to the folded conformation, the proposed method guides the protein into a region of higher energy. This increase is initially more steep ($0{\rightarrow}1500$) than in the later stages. Additionally, most of the finer-grained folding ($2500{\rightarrow}4000$) occurs with a high potential energy before settling into the lower-energy folded state. We notice that for the restricted folding time we use in our experiments ($5000\,\text{fs}$ vs $0.6\,\mu\text{s}$), the molecule does not end at the final configuration but reaches close to it as shown by the plot on pairwise distance. Furthermore, the learned policy network is able to transition through the high energy transition barrier in this restricted time. We do not encounter this for molecules with a shorter natural transition time (as illustrated by the potential energy of Alanine Dipeptide in fig. 2).

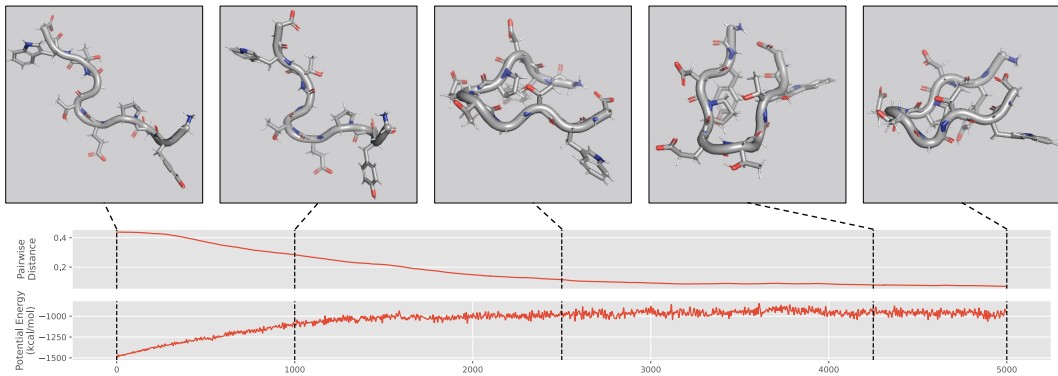

Figure 4: Visualization of the Chignolin folding process. *Top:* 5 stages of the folding process, *Middle:* Pairwise distance wrt to the target conformation of the molecule, *Bottom:* Potential Energy.

## 6   Discussion

In this work, we have proposed PIPS—a path integral stochastic optimal control method for the problem of molecular sampling transition paths using a bias potential. In contrast to prior work, PIPS does not require prespecifying CVs along which the system should be biased. We show the benefits of PIPS using three different molecular systems of varying sizes. In passing, we gave an introductory description of the problem of sampling transition paths and related it to the stochastic optimal control and the Schrodinger bridge problem. With this, we hope to not only have motivated our own work but also provided a starting point for future work consideration of this important problem by the machine learning community. For future work, we specifically note that the use of PIPS for CV discovery and the exploration of other approaches for specifying the target state, possibly using an ensemble of samples, is a promising direction as exemplified by our Polyproline experiment.

## Acknowledgements

We would like to thank Rianne van den Berg for their valuable feedback. Lars Holdijk is supported by the EPSRC Centre for Doctoral Training in Autonomous Intelligent Machines and Systems (EP/S024050/1).

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

## A   Proof theorem: SOC solves the BPTP problem

**Theorem A.1** (SOC solves the BPTP problem). *Given $\boldsymbol{x}_t = (\boldsymbol{r}_t, \boldsymbol{v}_t)^T$, $\boldsymbol{f}(\boldsymbol{x}_t) = (\boldsymbol{v}_t, \frac{-\nabla_{\boldsymbol{r}_t} U(\boldsymbol{r}_t)}{\boldsymbol{m}} - \gamma \boldsymbol{v}_t)^T$, $\boldsymbol{G}(\boldsymbol{x}_t) = (\boldsymbol{0}_{3n}, \mathbb{I}_{3n})^T$, $\boldsymbol{u}(\boldsymbol{x}_t) = \frac{-\nabla_{\boldsymbol{r}_t} b(\boldsymbol{r}_t, \boldsymbol{v}_t)}{\boldsymbol{m}}$, $\nu = 2\boldsymbol{m}\gamma k_B T$, and $\pi_0 = \pi_G$, such that the SOC dynamics (eq. (11)) describe the dynamics of the BPTP distribution $\pi^b$ (eq. (5)).*

*If we define $\varphi(\boldsymbol{x}_\tau) = -\lambda \log(\boldsymbol{1}_P(\boldsymbol{r}_\tau))$, $\boldsymbol{R} = \lambda \nu^{-1} = \lambda(2\boldsymbol{m}\gamma k_B T)^{-1}$ and assume $\boldsymbol{r}_0 \in R$, we have*

$$\arg\min_b \mathbb{E}_{\boldsymbol{x}_{0:\tau} \sim \pi^b}\left[C(\boldsymbol{x}_{0:\tau})\right] = \arg\min_b \mathbb{D}_{\mathsf{KL}}(\pi^b | \pi^*), \tag{17}$$

*where $\pi^*$ is the TP distribution.*

*Proof.* Let $\pi^b$ be the BPTP distribution as defined in eq. (7). Crucially, $\pi^b$ can be factored into a position and velocity component based on the conditional independence of $\boldsymbol{r}_{t+1}$ and $\boldsymbol{v}_{t+1}$ given $\boldsymbol{r}_t$ and $\boldsymbol{v}_t$, respectively, as

$$\pi^b(\boldsymbol{x}_{0:\tau}) = \pi^b_{\boldsymbol{r}}(\boldsymbol{x}_{0:\tau}) \cdot \pi^b_{\boldsymbol{v}}(\boldsymbol{x}_{0:\tau}) \tag{18}$$

with

$$\pi^b_{\boldsymbol{r}}(\boldsymbol{x}_{0:\tau}) = \prod_{t=0}^{\tau} \mathbb{1}_{[\boldsymbol{r}_{t+1} = \boldsymbol{r}_t + \boldsymbol{v}_t]}(\boldsymbol{r}_{t+1}) \tag{19}$$

$$\pi^b_{\boldsymbol{v}}(\boldsymbol{x}_{0:\tau}) = \prod_{t=0}^{\tau} \mathcal{N}(\boldsymbol{v}_{t+1} | \boldsymbol{\mu}_t, \Sigma_t). \tag{20}$$

where $\boldsymbol{\mu}_t = (\boldsymbol{v}_t \cdot \mathrm{d}t, \frac{-\nabla_{\boldsymbol{r}}\left(U(\boldsymbol{r}_t) + b(\boldsymbol{r}_t, \boldsymbol{v}_t)\right)}{\boldsymbol{m}} \cdot \mathrm{d}t - \gamma \boldsymbol{v}_t \cdot \mathrm{d}t)^T$ and $\Sigma = \mathrm{diag}(0, 2\boldsymbol{m}\gamma k_B T)$.

Now, if we define $\pi^0$ to be the BPTP distribution where no additional bias potential is applied, i.e. $b(\boldsymbol{r}_t, \boldsymbol{x}_t) = 0$ such that $\pi^0(\boldsymbol{x}_{0:\tau}) = \boldsymbol{1}_R(\boldsymbol{r}_0) \cdot \pi(\boldsymbol{x}_{0:\tau})$, we observe that the position component of the factorization are equal: $\pi^b_{\boldsymbol{r}}(\boldsymbol{x}_{0:\tau}) = \pi^0_{\boldsymbol{r}}(\boldsymbol{x}_{0:\tau})$.

Following, we use Girsanov's [Cameron and Martin, 1944] theorem to relate $\pi^b_{\boldsymbol{v}}(\boldsymbol{x}_{0:\tau})$ and $\pi^0_{\boldsymbol{v}}(\boldsymbol{x}_{0:\tau})$ as

$$\pi^b_{\boldsymbol{v}}(\boldsymbol{x}_{0:\tau}) = \pi^0_{\boldsymbol{v}}(\boldsymbol{x}_{0:\tau}) \cdot \exp\left(\frac{1}{\lambda} \sum_{t=0}^{\tau-1} \frac{1}{2} \boldsymbol{u}(\boldsymbol{x}_t)^T \boldsymbol{R}\boldsymbol{u}(\boldsymbol{x}_t) + \boldsymbol{u}(\boldsymbol{x}_t)^T \boldsymbol{R}\varepsilon_t\right) \tag{21}$$

where $\varepsilon = \boldsymbol{G}^{-1}(\boldsymbol{x}_t)(\mathrm{d}\boldsymbol{x} - \boldsymbol{f}(\boldsymbol{x}_t)\,\mathrm{d}t) - \boldsymbol{u}(\boldsymbol{x}_t)$. Which, given the previously established equality between the velocity components of the BPTP factorization, gives us

$$\log \frac{\pi^b(\boldsymbol{x}_{0:\tau})}{\pi^0(\boldsymbol{x}_{0:\tau})} = \frac{1}{\lambda} \sum_{t=0}^{\tau-1} \frac{1}{2} \boldsymbol{u}(\boldsymbol{x}_t)^T \boldsymbol{R}\boldsymbol{u}(\boldsymbol{x}_t) + \boldsymbol{u}(\boldsymbol{x}_t)^T \boldsymbol{R}\varepsilon_t \tag{22}$$

where $\varepsilon = \boldsymbol{G}^{-1}(\boldsymbol{x}_t)(\mathrm{d}\boldsymbol{x} - \boldsymbol{f}(\boldsymbol{x}_t)\,\mathrm{d}t) - \boldsymbol{u}(\boldsymbol{x}_t)$.

This allows us to rewrite the control cost eq. (13) as

$$C(x_{0:\tau}) = \frac{1}{\lambda}\left(\varphi(\boldsymbol{x}_{0:\tau})\right) + \log \frac{\pi^b(\boldsymbol{x}_{0:\tau})}{\pi^0(\boldsymbol{x}_{0:\tau})} \tag{23}$$

Finally, this gives

$$\arg\min_b \mathbb{E}_{\boldsymbol{x}_{0:\tau}\sim\pi^b}\big[C(\boldsymbol{x}_{0:\tau})\big] = \arg\min_b \mathbb{E}_{\boldsymbol{x}_{0:\tau}\sim\pi^b}\Big[\frac{1}{\lambda}\Big(\varphi(\boldsymbol{x}_\tau)\Big) + \log\frac{\pi^b(\boldsymbol{x}_{0:\tau})}{\pi^0(\boldsymbol{x}_{0:\tau})}\Big] \tag{24}$$

$$= \arg\min_b \mathbb{E}_{\boldsymbol{x}_{0:\tau}\sim\pi^b}\Big[-\log(\mathbf{1}_P(\boldsymbol{r}_\tau)) + \log\frac{\pi^b(\boldsymbol{x}_{0:\tau})}{\pi^0(\boldsymbol{x}_{0:\tau})}\Big] \tag{25}$$

$$= \arg\min_b \mathbb{E}_{\boldsymbol{x}_{0:\tau}\sim\pi^b}\Big[\log\frac{\pi^b(\boldsymbol{x}_{0:\tau})}{\mathbf{1}_R(\boldsymbol{r}_\tau)\cdot\pi(\boldsymbol{x}_{0:\tau})\cdot\mathbf{1}_P(\boldsymbol{r}_\tau)}\Big] \tag{26}$$

$$= \arg\min_b \mathbb{E}_{\boldsymbol{x}_{0:\tau}\sim\pi^b}\Big[\log\frac{\pi^b(\boldsymbol{x}_{0:\tau})}{\pi^*(\boldsymbol{x}_{0:\tau})}\Big] \tag{27}$$

$$= \arg\min_b \mathbb{D}_{\mathsf{KL}}(\pi^b|\pi^*) \tag{28}$$

where $\pi^*$ is the TP distribution as defined in definition 1.

$\square$

# B  Algorithms

---

**Algorithm 1:** Training Policy $\boldsymbol{u}_\theta$

---

**Input:** $\boldsymbol{r}_0, \boldsymbol{r}_T$: *Initial and target molecular positions,*
$\quad\quad U(\cdot)$: *Potential Energy function,*
$\quad\quad \gamma$: *Langevin Friction,*
$\quad\quad \varphi(\cdot)$: *Terminal cost,*
$\quad\quad \boldsymbol{u}_\theta(\cdot,\cdot)$: *Initial parameterized policy,*
$\quad\quad N$: *Number of trajectories sampled per update,*
$\quad\quad \tau$: *Time horizon,*
$\quad\quad \nu$: *Variance of Brownian noise,*
$\quad\quad \boldsymbol{R}$: *Control cost matrix,*
$\quad\quad \mu$: *Learning rate,*
$\quad\quad \mathrm{d}t$: *Time discretization step*
**while** *not converged* **do**
$\quad\triangleright$ *Generate trajectories with current policy $\boldsymbol{u}_\theta$*
$\quad \lambda \leftarrow \boldsymbol{R}\nu$ ;
$\quad n \leftarrow 0$ ;
$\quad$**while** $n < N$ **do**
$\quad\quad\triangleright$ *Initialize initial trajectory state*
$\quad\quad (\boldsymbol{r}_{n,0}, \boldsymbol{v}_{n,0}, t) \leftarrow (\boldsymbol{r}_0, \mathbf{0}, 0)$;
$\quad\quad$**while** $t < (\tau/\mathrm{d}t)$ **do**
$\quad\quad\quad\triangleright$ *Sample Brownian noise and action*
$\quad\quad\quad \varepsilon_{n,t} \sim \mathcal{N}(0, \sqrt{2\boldsymbol{m}\gamma k_B T})$;
$\quad\quad\quad \hat{\varepsilon}_{n,t} \sim \mathcal{N}(0, \nu)$;
$\quad\quad\quad \boldsymbol{u}_{n,t} \leftarrow \boldsymbol{u}_\theta(\boldsymbol{r}_{n,t}, t)$;
$\quad\quad\quad\triangleright$ *Update positions and velocity*
$\quad\quad\quad \boldsymbol{r}_{n,t+1} \leftarrow \boldsymbol{r}_{n,t} + \boldsymbol{v}_{n,t}\cdot\mathrm{d}t$;
$\quad\quad\quad \boldsymbol{v}_{n,t+1} \leftarrow \boldsymbol{v}_{n,t} + \Big(\frac{-\nabla_{\boldsymbol{r}}U(\boldsymbol{r})}{\boldsymbol{m}} + \boldsymbol{u}_{n,t} - \gamma\boldsymbol{v} + \varepsilon_{n,t} + \hat{\varepsilon}_{n,t}\Big)\cdot\mathrm{d}t$;
$\quad\quad\quad t \leftarrow t + 1$;
$\quad\quad$**end**
$\quad\quad\triangleright$ *Determine trajectory cost and gradient*
$\quad\quad C_n \leftarrow \frac{1}{\lambda}(\varphi(\boldsymbol{r}_{n,\tau}) + \sum_{i=0}^{\tau}\boldsymbol{u}_{n,i}^T\boldsymbol{R}\boldsymbol{u}_{n,i} + \boldsymbol{u}_{n,i}^T\boldsymbol{R}\varepsilon_{n,i})$;
$\quad\quad \Delta\theta_n \leftarrow \exp(-C_n) + \sum_{i=0}^{\tau}\frac{\partial\boldsymbol{u}_{n,i}}{\partial\theta}\boldsymbol{R}\varepsilon_{n,i}$;
$\quad\quad n \leftarrow n + 1$ ;
$\quad$**end**
$\quad\triangleright$ *Determine gradient normalization and perform policy update*
$\quad \eta \leftarrow \sum_{i=0}^{N}\exp(-C_i)$;
$\quad \theta \leftarrow \theta + \frac{\mu}{\eta}\sum_{i=0}^{N}\Delta\theta_i$;
**end**

---

# C   Extension Experimental section

## C.1   OpenMM

**General setup:**   We use the Velocity Verlet with Velocity Randomization (VVVR) integrator [Sivak et al., 2013] within OpenMM at a temperature of $300\,\text{K}$ with a collision rate of $1.0\,\text{ps}^{-1}$. All code is implemented in Pytorch and ran on a single GPU (either an NVIDIA RTX3080 or RTX2080).

**Alanine Dipeptide:**   We use the amber 99sb-ildn force field [Lindorff-Larsen et al., 2010] without any solvent, a time-step of $1.0\,\text{fs}$ for the VVVR integrator and a cutoff of $1\,\text{nm}$ for the Particle Mesh Ewald (PME) method [Essmann et al., 1995]. The policy network for 15000 roll-outs with a time horizon of $500\,\text{fs}$ each consisting of 16 samples. A gradient update was made to the policy network after each roll-out with a learning rate of $10^{-5}$. The Brownian motion has a standard deviation of $0.1$.

**Polyproline Helix:**   We initialize OpenMM with the amber protein.ff14SBonlysc forcefield and gbn2 as the implicit solvent forcefield. The VVVR integrator had a timestep of $2.0\,\text{fs}$ and a cutoff of $5\,\text{nm}$ for PME. The proposed method was ran for a total of $10.000\,\text{fs}$ (resulting in 5,000 policy steps). The policy networks was trained over 500 rollouts with 25 samples each using a learning rate of $3 \times 10^{-5}$ and a standard deviation of $0.1$ for the Brownian motion.

**Chignolin:**   To sample transition paths between the folded and unfolded state of the Chignolin protein, we initialize OpenMM using the same forcefield and VVVR integrator as for Polyproline with the exception that we sample a new force from our policy network every $1.0\,\text{fs}$. We do this 5000 times for each rollout for a total time horizon of $5000\,\text{fs}$. The policy network is trained for 500 roll-outs of 16 samples with a learning rate of $10^{-4}$ and a standard deviation of $0.05$ for the Brownian motion.

## C.2   Alanine Dipeptide

### C.2.1   Discussion Baselines and Evaluation Metrics

**Metrics**   Three different metrics are used for the comparison covering multiple desiderata for the sampled transition trajectories. For each metric we report the score over 1000 trajectories with the exception of the *Molecular Dynamics without fixed timescale* baseline which is only ran until 10 trajectories are successfully generated.

*Expected Pairwise Distance (EPD)* The EPD measures the similarity between the final conformation in the trajectory and the target conformation taking into account the full 3D geometry of the molecule. Note that the expected pairwise distance for uncontrolled MD with the target as the starting conformation has a EPD of $2.25 \times 10^{-3}$. All trajectories with an EPD of less than this can thus be considered to transition the molecule within one standard deviation of the target distribution.

*Target Hit Percentage (THP):* The second metric under which we evaluate the proposed Transition Path Sampler measures the similarity of the final and target conformation in terms of the collective variables. The THP measures the percentage of generated trajectories/paths that reach the target state. As such, higher hit percentages are preferred. We determine a trajectory to have hit the target in CV space when $\phi$ and $\psi$ are both within $0.75$ of the target.

*Energy Transition Point (ETP):* The final metric looks at the potential energy of the transition point—the conformation in the trajectory with the highest potential energy. This directly evaluates the capability of the method to find the transition path that crosses the boundary at the lowest saddle point.

**Baselines**   We compare the proposed Transition Path Sampling method with extended Molecular Dynamics simulation using different time-scales and temperature points. As discussed earlier, there are currently no other methods available for Transition Path Sampling using the full 3D geometry of the molecules.

*Molecular Dynamics with fixed timescale:* This set of baselines is limited to the same timescale as the proposed Transition Path Sampler, 500 femtoseconds, but uses varying temperatures. With

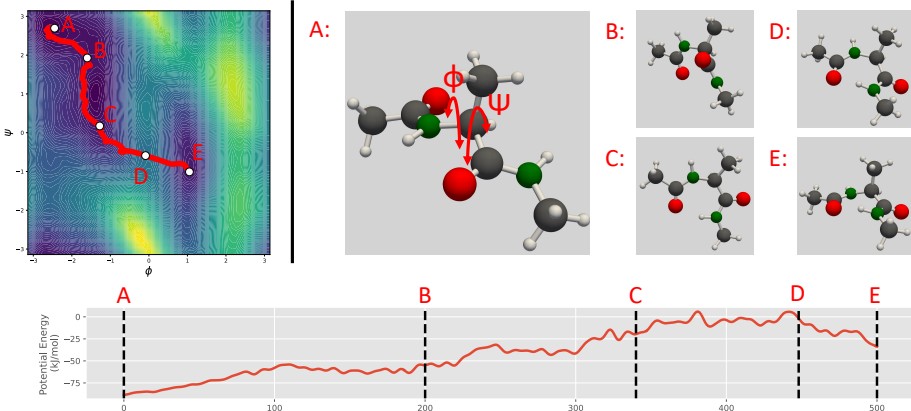

Figure 5: Visualization of a trajectory sampled with the proposed force prediction method. *Left:* The sampled trajectory projected on the free energy landscape of Alanine Dipeptide as a function of two CVs *Right:* Conformations along the sampled trajectory: A) starting conformation showing the CV dihedral angles, B-D) intermediate conformations with D being the highest energy point on the trajectory, and E) final conformation, which closely aligns with the target conformation. *Bottom:* Potential energy during transition. Letters represent the same configurations in the transition.

higher temperatures we should have a higher probability of crossing the barrier and hitting the target configuration.

*Molecule Dynamics without fixed timescales:* In contrast to the other set of baselines, the MD simulation for this set is not limited to 500 femtoseconds, but is instead ran until the target conformation is reached. We consider a trajectory to have reached its target if the following two conditions have been met: 1) the current conformation classifies as having hit the target under the conditions of the metric described above and 2) the current conformation is within one standard deviation of the target distributions mean.

By running the MD simulations until the target is reached we aim to gain intuition into the speed-up that it achieved by the fixed timescale of the proposed Transition Path Sampler.

### C.2.2 Additional results: Visualization Force Prediction

We observe that the force predicting policy has learned a different trajectory then the energy predicting model presented in the main body of the paper. While different, both of the trajectories pass the high energy barrier in a locally low point. Previous work on finding transition path has also observed that multiple viable paths can be found for Alanine Dipeptide [Hooft et al., 2021].