# 4 PIPS: Path Integral SOC for Path Sampling

Previously, we have seen how SOC solutions are also solutions for the BPTP problem. Using this insight, we will design a SOC approach to finding a parameterized bias potential $b_\theta$, that solves the BPTP problem. We refer to this method as PIPS: Path Integral Path Sampling. PIPS is an adaptation of the Path Integral Cross Entropy (PICE) [Kappen and Ruiz, 2016] method to the setting of sampling molecular transition paths where we have a single initial $R = \{\boldsymbol{r}_0^*\}$ and target $P = \{\boldsymbol{r}_\tau^*\}$ system.

**Background: Path Integral Cross Entropy** Kappen and Ruiz [2016] introduced the Path Integral Cross Entropy (PICE) method for solving Equation (12). The PICE method derives an explicit expression for the distribution $\pi_{\boldsymbol{u}^*}$ under optimal control $\boldsymbol{u}^*$ when $\lambda = \nu \boldsymbol{R}$ given by:

$$\pi^{\boldsymbol{u}^*} = \frac{1}{\eta(\boldsymbol{x}, t)} \pi^{\boldsymbol{u}}\big(\boldsymbol{x}_{0:\tau}\big) \exp(-C(\boldsymbol{x}_{0:\tau})) \tag{15}$$

where $\eta(\tau) = \mathbb{E}_{\boldsymbol{x}_{0:\tau} \sim \pi^0}[\exp(-\frac{1}{\lambda}\varphi(\boldsymbol{x}_\tau)]$ is the normalization constant. This establishes the optimal distribution $\pi^{\boldsymbol{u}^*}$ as a reweighing of any distribution induced by an arbitrary control $\boldsymbol{u}$.

PICE, subsequently, achieves this by minimizing the KL-divergence between the optimal controlled distribution $\pi^{\boldsymbol{u}^*}$ and a parameterized distribution $\pi^{\boldsymbol{u}_\theta}$ using gradient descent as follows:

$$\frac{\partial \mathbb{D}_{\mathsf{KL}}(\pi^{\boldsymbol{u}^*}|\pi^{\boldsymbol{u}_\theta})}{\partial \theta} = -\frac{1}{\eta} \mathbb{E}_{\boldsymbol{x}_{0:\tau} \sim \pi_{\boldsymbol{u}_\theta}}[\exp(-C(\boldsymbol{x}_{0:\tau}, \boldsymbol{u}_\theta)) \sum_{t=0}^{\tau} (\boldsymbol{R}\varepsilon_t \cdot \frac{\partial \boldsymbol{u}_\theta}{\partial \theta})] \qquad (16)$$

Similar to the optimal control in eq. (15), the gradient used to minimize the KL-divergence is found by reweighing for each sampled trajectory, $\boldsymbol{x}_{0:\tau}$, the gradient of the control policy $\boldsymbol{u}_\theta$ by the cost of the trajectory. See Algorithm 1 in the appendix for an algorithmic description of PICE.

### 4.1 Adaptations to PICE

In this section we will specify the adaptations made to the PICE algorithm to apply it to solve the BPTP problem for the molecular transition path setting.

**Smoothing the loss function** As shown in the previous section, when using the target loss $\varphi(\boldsymbol{x}_\tau) = -\lambda \log(\boldsymbol{1}_P(\boldsymbol{r}_\tau))$, SOC solves the BPTP problem. However, while optimal, this loss function is hard to use in the PICE optimization task as it is infinite for all $\boldsymbol{x}_{0:\tau}$ where $\boldsymbol{r}_\tau \neq \boldsymbol{r}_\tau^*$. As such, we instead use a smoothed version $\varphi(\boldsymbol{r}_t) = \exp \sum_{i,j}^{n} (d_{ij}(\boldsymbol{r}_t) - d_{ij}(\boldsymbol{r}_\tau))^2$ where $d_{ij}(\boldsymbol{r}_t) = \|(\boldsymbol{r}_t)_i - (\boldsymbol{r}_t)_j\|_2^2$. This exponentiated pairwise distance between the atoms is a commonly used distance metric [Shi et al., 2021] that is invariant to rotations and translations of the molecular system.

**Architectural considerations** The learnable component of PIPS is the bias potential $b$. However, as the BPTP dynamics show in eq. (5), instead of using the bias potential directly, MD depends on the *bias force* — the gradient of the bias potential $\boldsymbol{b}(\boldsymbol{r}, \boldsymbol{v}) = \nabla_{\boldsymbol{r}} b(\boldsymbol{r}, \boldsymbol{v}) \in \mathbb{R}^{3n}$. This consideration allows for two different modelling approaches for the bias force similar to the distinction between metadynamics and adaptive bias force discussed in section 2.3.1. One can either parameterise the bias force directly $\boldsymbol{b}(\boldsymbol{r}, \boldsymbol{v}) = \boldsymbol{b}_\theta(\boldsymbol{r}, \boldsymbol{v})$ or , alternatively, model $b_\theta(\boldsymbol{r}, \boldsymbol{

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

\big(\boldsymbol{x}_{0:\tau}\big) = \pi_{\boldsymbol{r}}^b\big(\boldsymbol{x}_{0:\tau}\big) \cdot \pi_{\boldsymbol{v}}^b\big(\boldsymbol{x}_{0:\tau}\big) \tag{18}$$

with

$$\pi_{\boldsymbol{r}}^b\big(\boldsymbol{x}_{0:\tau}\big) = \prod_{t=0}^{\tau} \mathbb{1}_{[\boldsymbol{r}_{t+1} = \boldsymbol{r}_t + \boldsymbol{v}_t]}(\boldsymbol{r}_{t+1}) \tag{19}$$

$$\pi_{\boldsymbol{v}}^b\big(\boldsymbol{x}_{0:\tau}\big) = \prod_{t=0}^{\tau} \mathcal{N}(\boldsymbol{v}_{t+1} | \boldsymbol{\mu}_t, \Sigma_t). \tag{20}$$

where $\boldsymbol{\mu}_t = (\boldsymbol{v}_t \cdot \mathrm{d}t, \frac{-\nabla_{\boldsymbol{r}}\big( U(\boldsymbol{r}_t) + b(\boldsymbol{r}_t, \boldsymbol{v}_t) \big)}{\boldsymbol{m}} \cdot \mathrm{d}t - \gamma \boldsymbol{v}_t \cdot \mathrm{d}t)^T$ and $\Sigma = \mathrm{diag}(0, 2\boldsymbol{m}\gamma k_B T)$.

Now, if we define $\pi^0$ to be the BPTP distribution where no additional bias potential is applied, i.e. $b(\boldsymbol{r}_t, \boldsymbol{x}_t) = 0$ such that $\pi^0(\boldsymbol{x}_{0:\tau}) = \boldsymbol{1}_R(\boldsymbol{r}_0) \cdot \pi(\boldsymbol{x}_{0:\tau})$, we observe that the position component of the factorization are equal: $\pi_{\boldsymbol{r}}^b\big(\boldsymbol{x}_{0:\tau}\big) = \pi_{\boldsymbol{r}}^0\big(\boldsymbol{x}_{0:\tau}\big)$.

Following, we use Girsanov's [Cameron and Martin, 1944] theorem to relate $\pi_{\boldsymbol{v}}^b\big(\boldsymbol{x}_{0:\tau}\big)$ and $\pi_{\boldsymbol{v}}^0\big(\boldsymbol{x}_{0:\tau}\big)$ as

$$\pi_{\boldsymbol{v}}^b\big(\boldsymbol{x}_{0:\tau}\big) = \pi_{\boldsymbol{v}}^0\big(\boldsymbol{x}_{0:\tau}\big) \cdot \exp\Big( \frac{1}{\lambda} \sum_{t=0}^{\tau-1} \frac{1}{2} \boldsymbol{u}(\boldsymbol{x}_t)^T \boldsymbol{R} \boldsymbol{u}(\boldsymbol{x}_t) + \boldsymbol{u}(\boldsymbol{x}_t)^T \boldsymbol{R} \boldsymbol{\varepsilon}_t \Big) \tag{21}$$

where $\boldsymbol{\varepsilon} = \boldsymbol{G}^{-1}(\boldsymbol{x}_t)(\mathrm{d}\boldsymbol{x} - \boldsymbol{f}(\boldsymbol{x}_t)\,\mathrm{d}t) - \boldsymbol{u}(\boldsymbol{x}_t)$. Which, given the previously established equality between the velocity components of the BPTP factorization, gives us

$$\log \frac{\pi^b\big(\boldsymbol{x}_{0:\tau}\big)}{\pi^0\big(\boldsymbol{x}_{0:\tau}\big)} = \frac{1}{\lambda} \sum_{t=0}^{\tau-1} \frac{1}{2} \boldsymbol{u}(\boldsymbol{x}_t)^T \boldsymbol{R} \boldsymbol{u}(\boldsymbol{x}_t) + \boldsymbol{u}(\boldsymbol{x}_t)^T \boldsymbol{R} \boldsymbol{\varepsilon}_t \tag{22}$$

where $\boldsymbol{\varepsilon} = \boldsymbol{G}^{-1}(\boldsymbol{x}_t)(\mathrm{d}\boldsymbol{x} - \boldsymbol{f}(\boldsymbol{x}_t)\,\mathrm{d}t) - \boldsymbol{u}(\boldsymbol{x}_t)$.

This allows us to rewrite the control cost eq. (13) as

$$C(x_{0:\tau}) = \frac{1}{\lambda}\Big( \varphi(\boldsymbol{x}_{0:\tau}) \Big) + \log \frac{\pi^b\big(\boldsymbol{x}_{0:\tau}\big)}{\pi^0\big(\boldsymbol{x}_{0:\tau}\big)} \tag{23}$$

Finally, this gives

$$\arg\min_b \mathbb{E}_{\boldsymbol{x}_{0:\tau}\sim\pi^b}\big[C(\boldsymbol{x}_{0:\tau})\big] = \arg\min_b \mathbb{E}_{\boldsymbol{x}_{0:\tau}\sim\pi^b}\Big[\frac{1}{\lambda}\Big(\varphi(\boldsymbol{x}_\tau)\Big) + \log\frac{\pi^b\big(\boldsymbol{x}_{0:\tau}\big)}{\pi^0\big(\boldsymbol{x}_{0:\tau}\big)}\Big] \tag{24}$$

$$= \arg\min_b \mathbb{E}_{\boldsymbol{x}_{0:\tau}\sim\pi^b}\Big[-\log(\boldsymbol{1}_P(\boldsymbol{r}_\tau)) + \log\frac{\pi^b\big(\boldsymbol{x}_{0:\tau}\big)}{\pi^0\big(\boldsymbol{x}_{0:\tau}\big)}\Big] \tag{25}$$

$$= \arg\min_b \mathbb{E}_{\boldsymbol{x}_{0:\tau}\sim\pi^b}\Big[\log\frac{\pi^b\big(\boldsymbol{x}_{0:\tau}\big)}{\boldsymbol{1}_R(\boldsymbol{r}_\tau)\cdot\pi\big(\boldsymbol{x}_{0:\tau}\big)\cdot\boldsymbol{1}_P(\boldsymbol{r}_\tau)}\Big] \tag{26}$$

$$= \arg\min_b \mathbb{E}_{\boldsymbol{x}_{0:\tau}\sim\pi^b}\Big[\log\frac{\pi^b\big(\boldsymbol{x}_{0:\tau}\big)}{\pi^*\big(\boldsymbol{x}_{0:\tau}\big)}\Big] \tag{27}$$

$$= \arg\min_b \mathbb{D}_{\mathsf{KL}}(\pi^b|\pi^*) \tag{28}$$

where $\pi^*$ is the TP distribution as defined in definition 1.

$\square$

# B  Algorithms

---

**Algorithm 1:** Training Policy $\boldsymbol{u}_\theta$

---

**Input:** $\boldsymbol{r}_0, \boldsymbol{r}_T$: *Initial and target molecular positions,*
  $\boldsymbol{U}(\cdot)$: *Potential Energy function,*
  $\gamma$: *Langevin Friction,*
  $\varphi(\cdot)$: *Terminal cost,*
  $\boldsymbol{u}_\theta(\cdot,\cdot)$: *Initial parameterized policy,*
  $N$: *Number of trajectories sampled per update,*
  $\tau$: *Time horizon,*
  $\nu$: *Variance of Brownian noise,*
  $\boldsymbol{R}$: *Control cost matrix,*
  $\mu$: *Learning rate,*
  $\mathrm{d}t$: *Time discretization step*
**while** *not converged* **do**
  $\triangleright$ *Generate trajectories with current policy* $\boldsymbol{u}_\theta$
  $\lambda \leftarrow \boldsymbol{R}\nu$ ;
  $n \leftarrow 0$ ;
  **while** $n < N$ **do**
    $\triangleright$ *Initialize initial trajectory state*