# OpenReview forum: "Stochastic Optimal Control for Collective Variable Free Sampling of Molecular Transition Paths"
_NeurIPS.cc/2023/Conference — NeurIPS 2023 poster_

### Official Review · Reviewer_vdgz · 2023-07-03

**Soundness:** 3 good
**Presentation:** 3 good
**Contribution:** 2 fair
**Rating:** 5
**Confidence:** 4

**Summary:**

The paper proposes an interesting method called PIPS for sampling transition paths between metastable states in molecular systems. These transitions are difficult to sample using standard Molecular Dynamics (MD) simulations due to high energy barriers. Traditional methods augment MD with a bias potential based on Collective Variables (CVs), but selecting appropriate CVs can be challenging and limiting for larger systems. PIPS instead does not rely on CVs and utilizes stochastic optimal control with neural network policies to efficiently learn the optimal bias potential sample molecular transitions. The method demonstrates success in generating low-energy transitions for several molecular systems.

**Strengths:**

1. Presentation: The technical part of the paper is very well-organized and easy to follow. Every component, like MD, optimal control, and their connections, are well explained to me.

2. Originality: The paper introduces a new ML method, PIPS, for sampling molecular transition paths without the need for Collective Variables. This offers a fresh perspective on solving the problem of high energy barriers in molecular dynamics simulations via stochastic optimal control.

3. Technical soundness: The paper establishes a formal relationship between the problem of sampling molecular transition paths and the stochastic optimal control. This theoretical foundation well motivates the credibility of the proposed method.

4. Generalizability: PIPS does not depend on CVs, making it applicable to many molecular systems where traditional methods may not be suitable,w without intense requirements for domain knowledge. This generalizability is particularly valuable as molecular systems are under exploration.

**Weaknesses:**

1. Transferability across all molecular systems. I list this critical limitation in the limitation section below.

2. ML contribution: One minor concern for accepting the paper to NeurIPS is that the paper is its contribution as an ML paper. ML technical parts in the paper, such as optimal control and practical neural network implementations, are all existing things. Therefore the ML contribution is not significant for the ML community.

3. Domain-specific impact: Continue from point 1, apart from ML contribution, I fully understand the value of the application paper for addressing a critical real-world problem. Then my second minor concern is that even the application scenario is a little niche. Topics such as MD acceleration and protein folding are definitely highly impactful, but the specific problem "Free Sampling of Molecular Transition Paths" sounds niche to me. Maybe provide a little more real-world impact and successful application examples in the paper?

4. I have several questions listed below.

**Questions:**

1. I may miss some details: is Schrodinger Bridge Problem really related to the method? I feel like the main method is just to use SOC to solve BPTP, and Sec 3.1 is kind of related but unnecessary. If so, I think it's unnecessary to incorporate these unnecessary contents that make the paper partially not coherent.

2. For the experiments, you train each model for a single molecular structure. Is it possible to train a model across several structures and make it generalizable to even other structures? This is the true pursuit of ML4Molecule studies. Just designing a method to solve a single structure's problem will be expensive and not widely useful in practice.

3. Why OpenMM is necessary for implementation? As the probabilistic formulation shown in equation 7, everything MD step is described as precise distributions. Then you can code the MD sampling and overcome the downstream mentioned in lines 237-242.

**Limitations:**

The main limitation in my mind is transferability to all systems: While PIPS shows promise in generating transition paths when trained on a single molecular system, is it possible to transfer the effectiveness to other or even all molecular systems? Or, for a weaker setting, transferable to arbitrary predefined initial and terminal states? If not, for even solving one structure, we need to train PIPS with extensive MD samplings, which might be too expensive.

---

> ### Author Rebuttal · Authors · 2023-08-09
>
> We thank the reviewer for their detailed comments. We appreciate the reviewers comments regarding the clarity and originality of the work.
>
> We have addressed the reviewer's concerns in detail below. Where possible, we have clustered related questions.
>
> We are looking forward to discussing further.
>
> ----
>
> >  ML technical parts in the paper, such as optimal control and practical neural network implementations, are all existing things. Therefore the ML contribution is not significant for the ML community.
>
> We appreciate that the reviewer flags the technical ML contribution as minor concern. We strongly believe that the introduction of CV-free sampling of molecular transition path as a problem domain for the ML community will lead to interesting technical ML innovations down the line as well.
>
> Additionally, we would also like to note that NeurIPS specifically welcomes application papers in its calls for papers.
>
> > Topics such as MD acceleration and protein folding are definitely highly impactful, but the specific problem "Free Sampling of Molecular Transition Paths" sounds niche to me. Maybe provide a little more real-world impact and successful application examples in the paper?
>
> We will make sure to add more real-world context to the introduction of our paper. In short, one can consider the problem of sampling molecular transition paths as an extension of MD acceleration. Even with current ML-potential based methods, the timescales required to sample transition within interesting systems is often still infeasible. Enhanced sampling is therefore typically a necessary requirement.
>
> The topic of protein folding is an application domain of this. Even with ML-potentials, MD simulation time is too short to be able to observe the folding process. As such, methods such as AlphaFold rely on one shot prediction of possible folded states. This however gives limited information on the transition process itself. Determining properties such as folding time or studying the folding mechanism itself requires enhanced sampling.
>
> > is Schrodinger Bridge Problem really related to the method? I feel like the main method is just to use SOC to solve BPTP, and Sec 3.1 is kind of related but unnecessary. If so, I think it's unnecessary to incorporate these unnecessary contents that make the paper partially not coherent.
>
> The connection with the Schrödinger Bridge problem is not required for establishing the connection between PIPS and the BPTP problem. However, we do believe that formalizing the relation between the SB problem and the BPTP problem is useful to inspire more work in this area.
>
> Based on discussion with other reviewers, we will consider moving this section to the appendix. We would greatly appreciate your feedback on whether or not you believe this is necessary.
>
> > Is it possible to train a model across several structures and make it generalizable to even other structures? This is the true pursuit of ML4Molecule studies.
>
> > While PIPS shows promise in generating transition paths when trained on a single molecular system, is it possible to transfer the effectiveness to other or even all molecular systems? Or, for a weaker setting, transferable to arbitrary predefined initial and terminal states?
>
> Thank you for this interesting question. Developing generalized policies is something we hope we, or others, can explore in future work. Both for generalizing across molecular systems and predefined initial and terminal states, we do expect this to be possible when the transitions rely on similar mechanisms (eg. a chain of rotating dihedral angles). Based on experience with CV-dependent methods, we foresee that this is a reasonable assumption. Furthermore, given a policy with sufficient expressive power, we expect that this restriction can also be lifted.
>
> However, we do believe that the study of a single molecular transition path can in itself already be worthwhile in domains such as drug discovery.
>
> > Why OpenMM is necessary for implementation? As the probabilistic formulation shown in equation 7, everything MD step is described as precise distributions.
>
> While OpenMM is not necessary for the implementation of PIPS, having access to a general purpose MD simulation framework significantly simplifies the computation of the Potential energy (and the related force) used in eq. 2.

---

### Official Review · Reviewer_VgsC · 2023-07-05

**Soundness:** 3 good
**Presentation:** 3 good
**Contribution:** 4 excellent
**Rating:** 7
**Confidence:** 4

**Summary:**

This paper addresses the task of sampling transition paths between metastable states in molecular systems. This problem is particularly pronounced when dealing with complex systems, where traditional methods often fall short due to the need for chemical intuition and the limitations in scalability. To overcome these obstacles, the authors propose a novel approach called path Integral
stochastic optimal control (PIPS). Unlike previous methods, PIPS does not rely on the explicit selection of collective variables (CVs), making it applicable to larger systems without sacrificing accuracy.


**Strengths:**

The paper provides a novel perspective and approach to sampling transition paths by leveraging stochastic optimal control theory and has already influenced a series of recent works. In particular, phrasing the problem as Schrödinger bridge (SB) problem is interesting and creative, as it presents a new application for optimal control as well as a clever parameterization of molecular dynamics frameworks.

**Weaknesses:**

Of course, the theoretical foundation utilized in this work is well-established in other fields. While this can be considered a weakness, I feel this work still provides a valuable contribution to the machine learning community.

The experiments are conducted on three molecular systems, i.e., alanine dipeptide, polyproline, and chignolin. While rather simple systems, I believe, these are sufficient illustrative examples to serve as an evaluation of this work. Could the authors still elaborate on how and why no larger molecules are considered?

For me, a core limitation is the lack of baselines, i.e., how do classical SBs or diffusion SBs such as [1] and [2] in this framework?

Is there any way to generalize to unseen molecules? Are there appropriate datasets available to tackle such tasks? What would be the necessary steps to achieve something like this?


[1] Tianrong Chen, Guan-Horng Liu, and Evangelos A. Theodorou. "Likelihood training of Schrödinger bridge using forward-backward SDEs theory." International Conference on Learning Representations (ICLR), 2022.

[2] Valentin De Bortoli, et al. "Diffusion Schrödinger bridge with applications to score-based generative modeling." Advances in Neural Information Processing Systems 34, 2021.

**Questions:**

The paper is very well structured and presented. No further questions on my end. I would classify myself as an informed reader and thus cannot access properly how intuitive the introduction is for readers with little background in control theory.

**Limitations:**

See weaknesses.

---

> ### Author Rebuttal · Authors · 2023-08-09
>
> We thank the reviewer for taking the time to review our work. We greatly appreciate the reviewers comments regarding the work's novelty and creativeness.
>
> We have the reviewer's concerns below in more detail. We look forward to further discussing our work.
>
> ---
>
> > The experiments are conducted on three molecular systems, i.e., alanine dipeptide, polyproline, and chignolin. Could the authors still elaborate on how and why no larger molecules are considered?
>
> We believe that by focussing on smaller systems, we can provide a more intuitive evaluation of the method. Larger systems have longer transition paths that contain a larger number of vital steps. Visualizing and interpreting these steps is non-trivial.
>
> Additionally, due to the complexity of MD simulation itself, scaling to larger systems requires additional engineering effort and compute. We would like to note that this is a limitation of MD (which requires multiple GPUs for modelling larger systems) and not PIPS.
>
> > For me, a core limitation is the lack of baselines, i.e., how do classical SBs or diffusion SBs such as [1] and [2] in this framework?
>
> Classical SBs or diffusion SBs are also appropriate approaches for modelling the Transition Path (TP) distribution, as exemplified by Thereom 3.1. However, in contrast to PIPS, these methods do not rely on a bias potential for approximating the TP distribution. This has some implications for downstream applications that we deemed outside the scope of this work.
>
> > Is there any way to generalize to unseen molecules? Are there appropriate datasets available to tackle such tasks? What would be the necessary steps to achieve something like this?
>
> Thank you for this very interesting question. Developing generalized policies is something we hope we, or others, can explore in future work. We expect that this is possible, but do foresee some roadblock. Most notably, we expect that the generalizability only extends to system with similar transition mechanisms (e.g. a series of dihedral angle changes). One would thus need to be able to construct a database of systems that share similar transition mechanisms. However, potentially this restriction can be alleviated with sufficiently expressive models.

---

> > ### Comment · Reviewer_VgsC · 2023-08-10
> > **Thank You for the Rebuttal**
> >
> > Thank you for you rebuttal. I will keep my score.

---

### Official Review · Reviewer_KPqs · 2023-07-11

**Soundness:** 3 good
**Presentation:** 3 good
**Contribution:** 3 good
**Rating:** 7
**Confidence:** 4

**Summary:**

This manuscript addresses the molecular transition pathway sampling problem, which is a fundamental and significant topic in computational chemistry/biology research. Specifically, the authors first made connections between transition path (TP) sampling, the Schrödinger Bridge Problem (SBP), and Stochastic Optimal Control (SOC), where the theoretical formulations are clearly described to facilitate understanding. Then, they proposed PIPS, a modified version of the PICE algorithm tailored to sample molecular transition paths without explicitly specifying the Collective Variables (CVs). It has been shown that one could learn a neural network bias potential (or force) policy to find the optimal transition paths in a few molecular systems, bypassing the need to use CVs or run long-time MD simulations.

I find this work interesting and enjoyed reading it. However, please see below for some of my concerns and questions.

**Strengths:**

- The research topic (i.e., molecular transition path sampling, rare-event sampling) is quite significant and has garnered much attention over the years. This work provides a novel perspective towards more effectively and efficiently identifying molecular conformation changes between metastable states.
- The paper is clearly written and the main ideas are easy to follow.
- I would say this work is inspirational and has the potential to lead to more follow-up work in this direction.

**Weaknesses:**

- The theoretical part of this work has already been developed for a while (i.e., Kappen and Ruiz, 2016). This work is an application of existing theoretical tools (with some modifications) to solve a particularly important biological problem. It is not necessarily a *weakness* yet I just put this comment here.
- While making connections between TP, BSP, and SOC, the transition into adopting the PISOC (line 177) is a little bit abrupt.
- The policy network u has not been discussed much, i.e., is MLP+act_fn a good universal solution to any molecular system under the PIPS framework?

**Questions:**

- For each molecular system, a new policy $\boldsymbol{u}$ needs to be trained. Therefore, is it correct that one needs to perform some hyperparameter tuning (i.e., Algorithm 1 in Appendix B) for each new system?
- Follow-up, is it possible to develop a generalized policy network across different molecular systems?
- While PIPS closely follow the PICE framework, I was wondering if there are other established methods also suitable for solving the TP sampling problem? It would be great if the authors could add some discussions about it.
- What is the impact of temperature on PIPS performance? In Table 1, PIPS is only run at 300K.
- The reported performance used MLP + ReLU activation, what is the impact of different network architectures? Some ablation studies would be appreciated.
- Line 326, while 5000 fs is much shorter compared with 0.6 us, why is the final configuration still close to the ground truth? Under the current statement, what happens between 5000 fs and 0.6 us?
- Minor, line 86, Canonical *or* NVT?
- Minor, line 322, "from its low energy unfolded state", do you mean the unfolded state is metastable, yet it has higher energy than the folded conformation?



**Limitations:**

The authors discussed PIPS limitations in the final Discussion section.

---

> ### Author Rebuttal · Authors · 2023-08-09
>
> We thank the reviewer for their time and feedback. We appreciate that the reviewer has found our work to be significant, clearly written, and inspirational.
>
> We will provide detailed comments to the reviewer's major remarks below. Where appropriate, we have grouped comments together. Minor remarks will be addressed in the next version of the manuscript.
>
> We look forward to further discussing our work.
>
> -------
>
> > The theoretical part of this work has already been developed for a while (i.e., Kappen and Ruiz, 2016). This work is an application of existing theoretical tools (with some modifications) to solve a particularly important biological problem. It is not necessarily a _weakness_ yet I just put this comment here.
>
> This is correct, and we appreciate that the reviewer acknowledges that his is not necessarily a weakness. We would like to add that NeurIPS specifically welcomes application papers in its call for papers.
>
> > While making connections between TP, BSP, and SOC, the transition into adopting the PISOC (line 177) is a little bit abrupt.
>
> We will address this in future versions by providing more background information on PISOC and providing some forward reference to Theorem 3.2. This should clarify why this specific branch of SOC is suitable for developing the PIPS framework.
>
> > The policy network u has not been discussed much, i.e., is MLP+act_fn a good universal solution to any molecular system under the PIPS framework?
>
> > The reported performance used MLP + ReLU activation, what is the impact of different network architectures? Some ablation studies would be appreciated.
>
> Thank you for your question. We expect that more physically aligned model architectures will allow for scaling PIPS to larger systems. Specifically, we believe that exploring equivariant, possibly graph based, architectures is an interesting direction. A short discussion of this is already included in the manuscript (l. 227- l. 229), but we will extend on this further.
>
> As the focus of this work is on showing that CV-free transition path sampling with a bias potential is feasible, we believe that providing an ablation study for different architectures is beyond its scope.
>
> > For each molecular system, a new policy � needs to be trained. Therefore, is it correct that one needs to perform some hyperparameter tuning (i.e., Algorithm 1 in Appendix B) for each new system?
>
> > Follow-up, is it possible to develop a generalized policy network across different molecular systems?
>
> This is a very interesting question. Currently, some hyperparameter tuning is required when switching between systems (learning rate, number of layers, etc. ). From experience, however, we find that it is rather straightforward to choose appropriate parameter settings based on system size.
>
> Developing generalized policies is something we hope we, or others, can explore in future work. We expect this to be possible.
>
> > While PIPS closely follow the PICE framework, I was wondering if there are other established methods also suitable for solving the TP sampling problem? It would be great if the authors could add some discussions about it.
>
> We expect that any established method for solving the Schrödinger Bridge problem can be applied to enhance the sampling of molecular transition paths.
>
> > What is the impact of temperature on PIPS performance? In Table 1, PIPS is only run at 300K.
>
> The temperature used for sampling the trajectories using PIPS is a function of the system itself and not a hyperparameter choice. By changing the temperature, the variance of the SDE in eq. 2 is changed.
>
> > Line 326, while 5000 fs is much shorter compared with 0.6 us, why is the final configuration still close to the ground truth? Under the current statement, what happens between 5000 fs and 0.6 us?
>
> Molecular transition paths are by definition not necessarily linear transition between point A and B, but is instead a very chaotic transition where, due to the stochasticity, the trajectory might backtrack at times before moving towards the target again. PIPS on the other hand enforces the transition to reach the target within a certain time frame (5000fs), which limits backtracking.

---

> > ### Comment · Reviewer_KPqs · 2023-08-17
> >
> > Thanks for your response, I have updated my score. I hope to see more follow-up research based on your work in the near future!

---

### Official Review · Reviewer_isij · 2023-07-26

**Soundness:** 2 fair
**Presentation:** 2 fair
**Contribution:** 2 fair
**Rating:** 3
**Confidence:** 3

**Summary:**

Molecular dynamics simulations yield trajectories which show how a system develops from a starting configuration to a final configuration over time.
Given a system with more than one (meta-)stable state, sampling trajectories which include transitions from one (meta-)stable state to another one might occur sparsely due to high energy barriers.
A solution is biasing the trajectory sampling procedure towards the ones which include transitions.
While classical approaches rely on Collective Variables (CV) to bias the sampling procedure, the authors present a CV-free method by learning an MLP, which takes the system configuration as inputs and biases the trajectory sampling procedure towards those who include transitions.
The authors recap that the problem of sampling these paths can be solved by stochastic optimal control theory and that related work suggested the PICE algorithm to find a way to control the path sampling.
They adapt the PICE algorithm and use the adaption - called PIPS - in their work.
PIPS allows to control the path sampling by introducing a learnable bias module, which is the MLP mentioned above.
The authors consider two versions of PIPS by modelling a) the bias potential and b) the bias force.
The authors apply their method to three different molecular systems.

**Strengths:**

(S1) **significance**: Being able to sample trajectories of molecular systems which include transitions is a significant problem in physics. Also, exploration of CV-free methods is a significant research direction for molecular dynamics related fields.

(S2) **originality**: Applying the adaption of PICE to molecular systems is novel. Also the comparison of the two versions of PIPS (a) and b)) includes some novelty. The smoothing of the loss function is introduced newly and intuitively seems to be an important detail for succesfully ML model training.

(S3) **quality**:
- Code is given. The illustration notebooks - e.g. ExampleAlanine.ipynb - helps to understand what was done and give insights to model training and path sampling.
- The section in which connections between transition path sampling, the Schrödinger bridge problem, and stochastic control theory are summarized and described is very informative and well understandable.

(S4) **clarity**: The problem setting and the introduction to molecular dynamics and potentially occurring problems in sampling transition paths under the presence of high energy barriers are written clearly and very well. Section 3 is well structured and explains very well why SOC theory is useful for the given problem setting.

**Weaknesses:**

(W1) **significance**:
- Applying ML models to molecular systems to sample rare events is already done here [1]. Notably, the input to the model described in [1] is CV-free since Cartesian coordinates of the atoms are fed into the model.
- While significance is given from a physics point of view, there is hardly any core ML-related novelty. The ML-related part of the manuscript is learning the bias function with a standard MLP.

(W2) **originality**: While (S2) is given, the connection between biasing path sampling and SOC is already described by Kappen and Ruiz. The adaption of PICE to derive the new method PIPS just seems to include a smoothing of the loss function. So it seems like the major part of the method is re-using PICE.

(W3) **quality**: The experimental section is flawed because of missing baselines and missing error bars and statistical tests:
- All performance values in Table 1 are reported without error bars. Also statistical tests are missing.
- Variation across training reruns is missing
- [1] is neglected
- CV-based baselines are not included but would be desired since it is unclear how PIPS performs compared to methods which use CVs.

(W4) **clarity**:
- Sparse method description (PIPS): Having described very well where PICE comes from and that PICE is able to provide a solution to the transition path sampling problem, the manuscript lacks information about the presented method (PIPS). E.g., pseudo code for training and sampling would have been helpful.
    - Additional confusion comes from the fact that the authors describe first that the PICE algorithm helps to find  a suitable choice of the control $\boldsymbol{u}_\theta$ but later they describe that the learnable component of PIPS is $b$ (or $\boldsymbol b$). Even though the relation between the control $\boldsymbol u$ and the bias potential is mentioned it remains unclear what the MLP exaclty models and how the model outcome exactly influnences the path sampling.
    - Even though some of the information might already be spread in the manuscript, the authors should think about providing (again) relevant information for PIPS in section 4. This also includes information about the exact loss function, training setup, and hyperparameter selection.
    - Theorem 3.2 shows that SOC solves the BPTP problem with the "standard" choice of $\varphi$ but the authors choose a different one for PIPS. The authors missed to include mathematical implication of changing $\varphi$ in terms of whether it is still guaranteed find a BPTP problem solution with SOC.
- Mathematical notation and formulas: The clarity is diminished due to unclear notation, e.g., $\pi _0$ (l. 171) vs $\pi^0$ (l. 191) and $\pi _{\boldsymbol u^*}$ (l. 201) vs $\pi ^{\boldsymbol u^*}$.
- Section 3 - contribution vs related work: It remains unclear which parts are taken from related work and which parts are newly developed. E.g, [2] includes already a lot of the findings presented in equations (9) - (14).


Minor points:
- l. 174 - typo: $\boldsymbol u(\boldsymbol x_t)$
- l. 184 - typo: (eq. (5))

[1] Sun, Lixin, et al. "Multitask machine learning of collective variables for enhanced sampling of rare events." Journal of Chemical Theory and Computation 18.4 (2022): 2341-2353.
[2] Kappen, Hilbert Johan, and Hans Christian Ruiz. "Adaptive importance sampling for control and inference." Journal of Statistical Physics 162 (2016): 1244-1266.


**Questions:**

1. The authors should include training reruns, report error bars, and perform statistical test for quantitative experiments
2. The authors should include SOTA CV-based methods and [1] as baselines and compare PIPS to them.
3. The author should include additional information about PIPS, e.g. a) what does the MLP predict exactly, b) what is the loss function, c) how does the training and sampling pipeline look like.
4. The authors should clarify unclear notation.
5. The authors should clarify which parts of section 3 are taken from related work and which parts are newly developed.


**Limitations:**

-

---

> ### Author Rebuttal · Authors · 2023-08-09
>
> We thank the reviewer for their detailed review of our work. We are happy to hear that the reviewer finds our work to be addressing a significant problem and in general well written and understandable.
>
> Below, we have included some detailed answers to your questions and comments. We have tried to group related comments together where appropriate.
>
> We are looking forward to further discussing our work.
>
> ---
>
> > Applying ML models ... already done here [1].
>
> Thank you for bringing this work to our attention. We will make sure to add the paper to the end of our related work section as an exploration of other ML-based approaches in this sphere.
>
> However, we would like to note that this work focusses on a problem setting that is significantly different from ours. The focus of the suggested work is not on CV-free enhanced sampling, but rather on the discovery of CVs for a given system. For the sampling of transition paths along these CVs, the paper in question still relies on umbrella sampling.
>
> The use of machine learning for enhanced sampling of molecular transitions using a CV-free bias potential has, to the best of our knowledge, not been done before.
>
> > While significance is ... any core ML-related novelty.
>
> The novelty of our work lies in the use of machine learning for learning a bias potential for enhancing the sampling of molecular transition paths that does not depend on Collective Variables. Bringing core ML-related novelty is not within the scope of our work.
>
> We would also like to note that NeurIPS specifically welcomes application papers in its calls for papers.
>
> > While (S2) is given, the ... described by Kappen and Ruiz.
>
> To the best of our knowledge, the formal connection between transition path sampling and SOC has not been introduced in prior work. The work of Kappen and Ruiz introduces the PICE algorithm, but does not discuss how this can be used for sampling molecular transition paths.
>
> > The adaption of PICE to ... smoothing of the loss function.
>
> This is correct. The acronym PIPS refers to the adaptation of PICE for the enhanced transition path sampling problem. This includes consideration such as the smoothing of the loss function and the incorporation within the OpenMM framework.
>
> > The experimental section .... and statistical tests:
>
> > The authors should include .... for quantitative experiments.
>
> > The authors should ... and compare PIPS to them.
>
> As we are introducing a new framework for enhanced sampling **without** using collective variables, the focus of our experimental section is on validating that PIPS can successfully generate transitions and that the generated transitions are physically correct. Thus, our experimental results are focussed on 2 main criteria for evaluating PIPS. Namely, with the bias potential trained without CV, running MD simulations should:
> 1. Generate rare trajectories that transition the system to the target state at a higher rate than without the bias potential. This is measured by the transition time, the THP and the EPD.
> 2. Generate rare trajectories that transition at a low point in the barrier separating the state. We validate this using the ETP and the Ramachandran plot.
> As our results show, PIPS is capable of frequently generating rare transitions that cross the barrier at a low saddle point in the free energy landscape.
>
> With this purpose of the experimental section in mind, we thus do not believe it to be useful to include comparisons against CV dependent methods, error bars and/or statistical tests. The goal of the experiments is not to show that PIPS is superior to CV dependent methods, it instead aims to demonstrate that bias potential enhanced transition path sampling is feasible without CVs.
>
> > CV-based baselines .... which use CVs.
>
> In addition to our general comment above, we would like to highlight that we do use CV-dependent methods to validate that our trajectories are physically meaningful. Namely, we compare our trajectories against the free-energy surface generated by meta-dynamics.
>
> > Even though some of the information .... and hyperparameter selection.
>
> > The author should include ....  training and sampling pipeline look like.
>
> We are unsure which additional information the reviewer would like to see included in section 4. We believe that section 5 is a more appropriate location to report experiment specific information, such as training setup and hyperparameter. The exact loss function is already specified in section 4 under “Smoothing the loss function”.
>
> Pseudocode for the training pipeline is provided in Appendix B. While it is unlikely that we will be able to fit the entire algorithm box in the main body of the paper, we will make sure that the reference stands out more. Similar Pseudocode for the sampling pipeline will be included in the next version.
>
> > It remains unclear which ... in equations (9) - (14).
>
> > The authors should clarify ... which parts are newly developed.
>
> We believe to have sufficiently addressed this in the paper. Eq. 10 and 14 show a novel formal relation between the BPTP problem and respectively the SB problem and SOC that was not discussed in [2]. Eq. 9, 11, 12, and 13 are paraphrased from numerous prior works. However, in all cases these equations are either part of a clear definition statement or included in a section labelled as “background”.
>
> > Mathematical notation and ....  $\pi_{u^*}$ (l. 201) vs $\pi^{u^*}$ .
>
> > l. 174 - typo: $u(x_t)$
> > l. 184 - typo: (eq. (5))
>
> > Additional confusion come ...  learnable component of PIPS is $b$ (or $b$).
>
> We will make the relation between the control policy and bias potential more clear in the updated version of the manuscript by stating the relationship between the two outside of Theorem 2. Additionally, we will address the small typos mistakes in the notation. To clarify, l.171 should read $\pi^0$ and l.201 should read $\pi^{u^*}$

---

> > ### Comment · Reviewer_isij · 2023-08-18
> >
> > I would like to thank the authors for their answers.
> >
> > Firstly, I'd really like to state that I do think that the author's work is relevant, especially for the chemoinformatics audience and related fields.
> >
> > Secondly, the **clarity** issue in terms of method description became obsolete since I overlooked the training algorithm in the Appendix. Thank your for pointing to it.
> >
> > However, I adhere to my rating since I still think weaknesses in the current manuscript version outweigh the strengths.
> >
> > ### Comments to the author's responses:
> >
> >
> > **Novelty**
> > > NeurIPS specifically welcomes application papers in its calls for papers
> >
> > This is correct. It was already acknowledged in (S1) and (S2). I nevertheless think W2 ("the connection between biasing path sampling and SOC is already described by Kappen and Ruiz") still holds (see Introduction of Adaptive Importance Sampling for Control and Inference).
> >
> > **Quality**
> > > we thus do not believe it to be useful to include comparisons against CV dependent methods, error bars and/or statistical tests.
> >
> > I disagree here. Especially, if a ML paper introduces a new field of application - which is here sampling molecular transition paths , it should:
> >
> > - compare the suggested approach, to methods which have been used so far for the problems shown in the experimental section. Here that includes CV-based methods as soon as there is any idea about what could be suitable as CVs.
> > - take care about setting a proper benchmark for future work. This only can be done by including error bars and statistical tests since any difference between two methods could arrise just by chance otherwise.
> >
> > **Clarity**
> > Method description and training details become very clear from Appendix B. I'd like to appologize for raising that point.

---

> > > ### Author Response · Authors · 2023-08-18
> > > **Further Clarifications**
> > >
> > > Thank you for your response. We are happy to hear that the reviewer found the provided algorithm box useful.
> > >
> > > **Regarding the remaining comments on novelty** We have reread the work of Kappen and Ruiz carefully, and can not find any mention of the problem of biased enhanced sampling in the context of molecular rare event sampling. The work by Kappen and Ruiz only introduces the mathematical framework of PICE, but does not focus on any application. Specifically, it does not formally show that it solves the BPTP problem.
> > >
> > > **Regarding the comparison to CV based methods** We would like to point the reviewer to the top left of figure 2, here we compare a CV-free generated transition from PIPS to the FES reconstructed using meta-dynamics. Meta-dynamics is the default CV based enhanced sampling method for biochemical applications and we use it in our work to validate our generated transition.
> > >
> > > **Regarding the reporting of error bars for future benchmarking** We would like to thank the reviewer for clarifying his concerns regarding the benchmarking for future work. We will add error bars for all quantitative evaluations presented in our work in the revised version to further facilitate future research in this area.
> > >
> > > With this we hope to have sufficiently addressed your concern. Please let us know if anything else needs further clarification.

---

> > > > ### Comment · Reviewer_isij · 2023-08-21
> > > >
> > > > > We [...] can not find any mention of the problem of biased enhanced sampling in the context of rare event sampling.
> > > >
> > > > This is true. The mentioned paper deals with the question of biasing path sampling though. Biasing molecular paths just requires a minor adaption.
> > > >
> > > > I see, that the adaption of a method to a new area includes some novelty. So does this paper - again, this was already acknowledged.
> > > > However, having the conference audience in mind, I still think the included novel parts (which are on the applied chemoinf. side) are not relevant enough for the NeuRIPS community.
> > > >
> > > > **Regarding CV based method comparison**:
> > > > Thank you for pointing to Figure 2, but this is not what I meant. I'd like to refer again to my last writing (section quality). Concretely, the authors should consider setting up a benchmark experiment which includes proper baselines and compared methods. In compared methods, CV-based methods should be included. The comparison should be done in a quantitative way which follows "best ML practices" by including reruns, and including error bars and statistical tests.
> > > >
> > > > I am glad to here that the authors are including error bars.

---

### Decision · Program_Chairs · 2023-09-21

**Decision:**

Accept (poster)

**Comment:**

There is some disagreement between reviewers here: two reviewers in favor of acceptance, one arguing to reject, and one borderline leaning accept. One of the reviewers raised their overall score following the rebuttal period.

I (the AC) am recommending acceptance, but the authors should make the proposed edits to the paper suggested in their response, including enhancements to the quantitative evaluation such as error bars.

For the final version, please do take into account the comments from all reviewers, including those regarding choice of quantitative baselines (including CV-dependent baselines where available).